# Characterization and structure-based protein engineering of a regiospecific saponin acetyltransferase from *Astragalus membranaceus*

Linlin Wang [1,4], Zhihui Jiang[2,4], Jiahe Zhang[1,4], Kuan Chen[1], Meng Zhang[1], Zilong Wang [1], Binju Wang [2] ✉, Min Ye [1,3] ✉ & Xue Qiao [1,3] ✉

Acetylation contributes to the bioactivity of numerous medicinally important natural products. However, little is known about the acetylation on sugar moieties. Here we report a saponin acetyltransferase from *Astragalus membranaceus*. AmAT7-3 is discovered through a stepwise gene mining approach and characterized as the xylose C3′/C4′-*O*-acetyltransferse of astragaloside IV (**1**). To elucidate its catalytic mechanism, complex crystal structures of AmAT7-3/**1** and AmAT7-3$_{A310G}$/**1** are obtained, which reveal a large active pocket decided by a specific sequence AADAG. Combining with QM/MM computation, the regiospecificity of AmAT7-3 is determined by sugar positioning modulated by surrounding amino acids including #A310 and #L290. Furthermore, a small mutant library is built using semi-rational design, where variants A310G and A310W are found to catalyze specific C3′-*O* and C4′-*O* acetylation, respectively. AmAT7-3 and its variants are also employed to acetylate other bioactive saponins. This work expands the understanding of saponin acetyltransferases, and provide efficient catalytic tools for saponin acetylation.

Saponins are widely distributed and structurally complex natural products with various bioactivities[1,2]. Acetylation is one of the most common modifications for saponins, and plays an important role in improving their bioactivity[2,3]. For examples, QS-7, an acetylated saponin (on fucose C-4) from *Quillaja saponaria*, is a favored anticancer and antiviral vaccine adjuvant candidate with negligible toxicity in mice[4]. Ginsenoside-Rs4 (acetylated on glucose C-6) strongly induced SK-HEP-1 cell death compared to ginsenoside-Rg5 (non-acetylated)[5]. Acetylation on oleanolic acid 3-*O*-arabinopyranoside (C-3 and C-4) improved the cytotoxicity on A431 cells by at least 10-fold[6]. Although per-*O*-chemical acylation strategies are highly developed[7,8], site-selective acetylation on sugar moieties is challenging due to the proximate relative activity of the hydroxyl groups. Previous attempts mainly focused on the protection of the anomeric group[9], and the acetylation on saponin molecules remains rare. In the chemical synthesis of QS-7, building an acetylated fucosyl residue required at least four steps, namely C3-OH alkylation, C2-OH silylation, acetylation, and C3 ether removal, with a yield of 47%[10]. Consequently, the synthesis of diverse naturally occurring acetylated saponins using chemical catalysts is challenging.

[1]State Key Laboratory of Natural and Biomimetic Drugs, School of Pharmaceutical Sciences, Peking University, 38 Xueyuan Road, Beijing 100191, China. [2]State Key Laboratory of Physical Chemistry of Solid Surfaces and Fujian Provincial Key Laboratory of Theoretical and Computational Chemistry, College of Chemistry and Chemical Engineering, Xiamen University, 361005 Xiamen, China. [3]Peking University-Yunnan Baiyao International Medical Research Center, 38 Xueyuan Road, Beijing 100191, China. [4]These authors contributed equally: Linlin Wang, Zhihui Jiang, Jiahe Zhang. ✉e-mail: wangbinju2018@xmu.edu.cn; yemin@bjmu.edu.cn; qiaoxue@bjmu.edu.cn

Saponin acetylation is most possibly catalyzed by acyl-transferases, specifically, glycoside acetyltransferases (ATs). Most functionally characterized enzymes in this group utilize aromatic acyl or malonyl groups as donors[11]. To date, only six acetyltransferases have been reported for glycosides, including four that catalyze flavonoid glucosyl C6′-OH (Vh3MAT1, Lp3MAT1, and CcAT1/2)[11] and two that modify tritpernoid fucosyl C4′-OH (QsACT1 and SOAP10)[4,12]. However, due to the rapid divergence of acyltransferases[13], the above enzymes showed high diversity either in our analysis (Fig. 1a) or in previous reports[11-14]. For instance, fucosyl ATs QsACT1 and SOAP10 share a sequence similarity of only 20.7%. On the other hand, though seven

acyltransferase crystal structures have been reported[15-21], none of them belong to glycoside ATs which accommodate saponins with larger size. Little is known about the catalytic mechanism and regiospecificity for the glycoside ATs. Therefore, it is challenging to discover ATs catalyzing the acetylation of saponins and to reveal their catalytic mechanism.

Astragalosides are the primary bioactive constituents of Astraga-lus Root, a popular herbal medicine worldwide deriving from *Astra-galus membranaceus*[22,23]. They possess a cycloartane-type skeleton and undergo diverse acetylation modifications on its C3-xylosyl group, resulting in the formation of compounds include astragaloside II

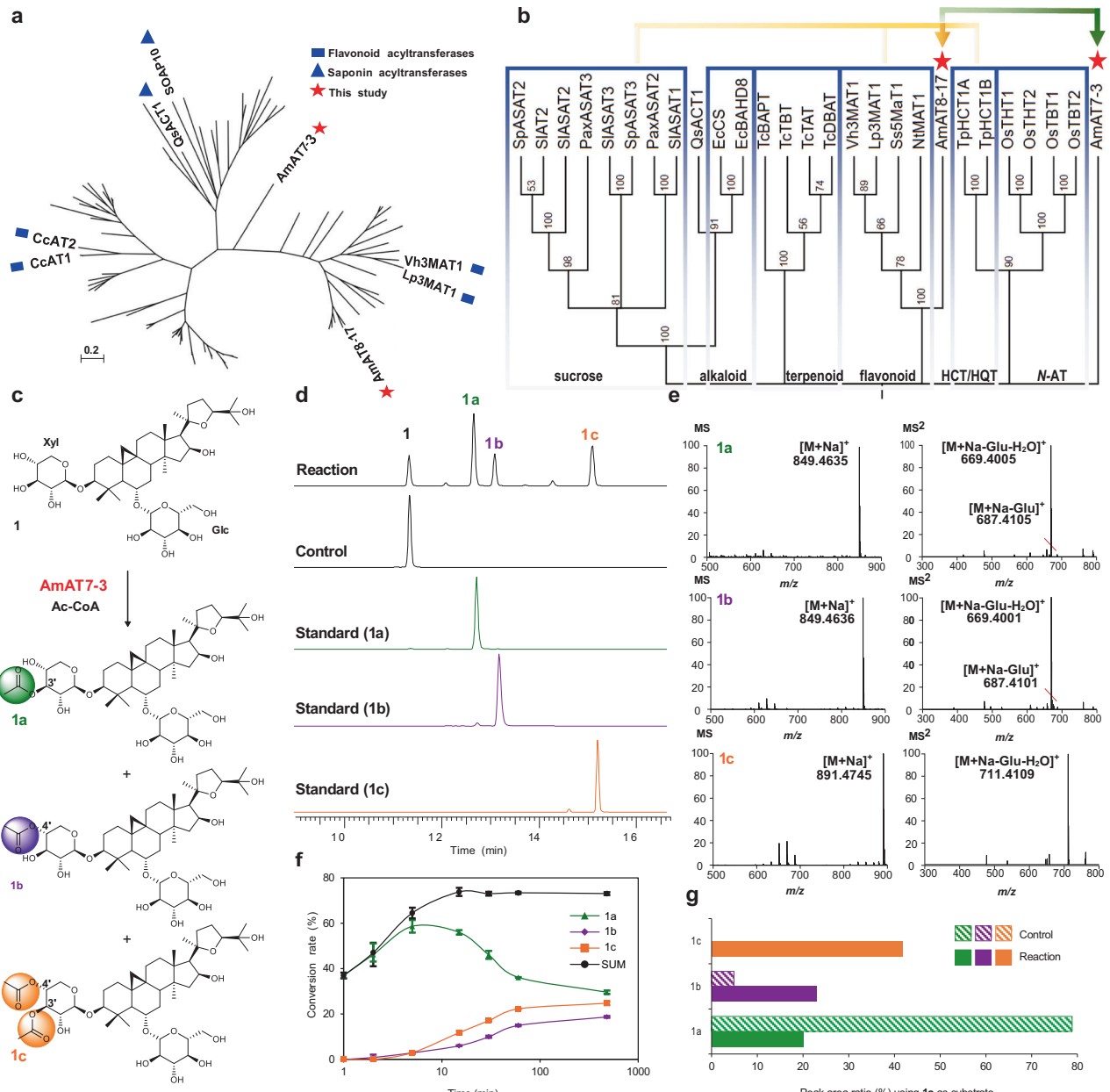

**Fig. 1 | Functional characterization of AmAT7-3. a** Maximum likelihood tree of acetyltransferases with diverse functions (Supplementary Data 1). **b** Maximum likelihood tree of AmAT7-3 with representative acyltransferases. Evolutionary analyses were conducted in MEGA7 with 1000 bootstrap replicates[44,45]. Yellow and green arrows indicate the first- and second-round gene screening, respectively. HCT/HQT, shikimic acid or quinic acid acyltransferase. **c** Acetylation of astragalo-side IV (**1**) catalyzed by AmAT7-3. **d** UPLC/MS total ion current chromatograms of the reaction, control, and the standard samples. **e** HR-MS and HR-MS/MS analysis of

products **1a, 1b, 1c** in positive ion mode. **f** Time-course of the enzymatic reaction of AmAT7-3 using **1** as the substrate. Data are presented in mean ± SD (*n* = 2 or 3 independent experiments). Source data are provided as a Source Data file. **g** Peak area ratio of **1a-1c** in the reaction mixture of AmAT7-3 using **1a** as the substrate. Glc, glucosyl group; Xyl, xylosyl group; Ac-CoA, acetyl-coenzyme A; Reaction: acety-lated reaction catalyzed by AmAT7-3; Control: boiled AmAT7-3 was used in the reaction mixture.

(**1d**, C2'-*O*-Ac), isoastragaloside II (**1a**, C3'-*O*-Ac), cyclocephaloside II (**1b**, C4'-*O*-Ac), astragaloside I (**1e**, C2',3'-*O*-Ac), isoastragaloside I (C2',4'-*O*-Ac), and acetylastragaloside I (C2',3',4'-*O*-Ac). Among them, astragalosides I and II are the most abundant saponins in Astragalus Root[24]. Astragalosides possess various bioactivities such as cardio-protective, neuroprotection and immunomodulatory effects, while the number and position of acetylation contributed largely to their bioactivity[25,26]. Due to the chemical complexity of Astragalus Root, it requires extensive purification to obtain these astragalosides from an extract containing at least 189 compounds[22]. Thus, it is necessary to study the biosynthetic route of astragalosides to expand the source of these saponins. Although our previous study had identified the scaffold-forming oxidosqualene cyclase and four glycosyl-transferases in the biosynthesis pathway[27–29], the downstream acetylation steps remain unclear. Based on the specific substrate (triterpenoid C3-xylose) and reaction sites (C2'/C3'/C4'), ATs that make astragalosides should be remarkably different from previously reported glycosyl ATs.

This work focuses on discovering and elucidating the catalytic mechanism of saponin acetyltransferases using Astragalus Roots as the plant material. Firstly, a saponin acetyltransferase, AmAT7-3, is discovered through a stepwise transcriptome analysis and biochemically characterized. Secondly, the complex crystal structure of AmAT7-3 (PDB: 8H8I) is obtained to gain insights into the catalytic mechanism, and semi-rational design is employed to engineer AmAT7-3 into regioselective enzymes. Complex crystal structure of the engineered variant AmAT7-3$_{A310G}$ (PDB: 8HBT) is also resolved. Finally, AmAT7-3 and its variants are used to modify medicinally important saponin molecules, thereby enhancing the structural diversity of natural products. The results expand the understanding of protein sequence, catalytic mechanism and protein engineering of saponin acetyltransferases, and provide efficient catalytic tools for glycoside acetylation.

## Results and discussion

### Gene screening and functional characterization

Due to the sequence diversity of glycoside acetyltransferases, a stepwise BLAST search was employed to screen the astragaloside acetyltransferases. Firstly, we selected nine ATs that modify six-membered rings as the templates (Supplementary Table 1). Their acyl acceptors include flavonoid glycosides, shikimic acid, and sucrose. A BLASTn search in the transcriptome of *A. membranaceus* (SRR923811) resulted 16 candidate genes ($e < 10^{-5}$) containing the conserved motifs HXXXD and DFGWG. Subsequently, these candidate genes were cloned into pET-28a (+) vectors, and the proteins were expressed in *Escherichia coli* (*E. coli*) (Supplementary Table 2). The function of the purified recombinant proteins was characterized in a mixture including 30 μg protein, 0.1 mM astragaloside IV (**1**), 0.5 mM acetyl-CoA and 0.5 mM dithiothreitol in 50 mM Na$_2$HPO$_4$-NaH$_2$PO$_4$ buffer (pH 6.0, 100 μL). The reaction was conducted in 30 °C for 30 min. Among these candidates, AmAT8-17 (GenBank accession No. OQ915518) demonstrate acetylation activity at the C4'-OH position of astragaloside IV (**1**) with a 25% conversion rate (Supplementary Fig. 1). However, aside from its poor activity, AmAT8-17 is unable to catalyze C2'-*O* or C3'-*O* acetylation, which are the primary acylation modification sites of astragalosides. Thus we used AmAT8-17 as the template for another BLASTn search, which yielded 10 additional candidate genes. Among them, a candidate, AmAT7-3 (ON804888), was found to modify astragaloside IV (**1**), producing three less polar peaks (**1a**, **1b**, **1c**) as shown in Fig. 1. The protein sequence of AmAT7-3 did not cluster with other acyl-transferases and formed a relatively distinct branch, indicating its high sequence specificity (Fig. 1a, b). The distribution of AmAT7-3, AmAT8-17, and the six known glycoside ATs explained the challenges in predicting the function of an acetyltransferase solely based on its protein sequence.

The mass spectra of **1a** and **1b** showed an [M+Na]$^+$ ion at 849.46 Da, which was 42.01 Da greater than that of compound **1**. The [M+Na]$^+$ ion of **1c** was 84.02 Da greater than that of **1**. These results indicated **1a**/**1b** as mono-acetylated products while **1c** as a di-acetylated product (Fig. 1c–e). The MS/MS spectra of **1** and all three products showed a neutral loss of 180.06 Da, which was consistent with a glucosyl group and a water molecule, indicating that acylation occurred on the xylosyl group. Structures of the products were further confirmed by comparing with reference standards. Products **1a**, **1b**, and **1c** were identified as C3'-*O*, C4'-*O*, and C3', C4'-*O* acetylated astragaloside IV, respectively (Fig. 1).

AmAT7-3 displayed its highest activity at 30 °C for 30 min at pH 6.0 (50 mM Na$_2$HPO$_4$-NaH$_2$PO$_4$ buffer), and it was independent from divalent metal ions (Supplementary Fig. 2). The in vitro conversion rate of AmAT7-3 reached 85.3%, which was higher than that of a number of previously characterized acyltransferases[11]. Since AmAT7-3 could catalyze the acetylation at different sites (C3' and C4'), we conducted a time-course experiment of 480 min to determine its preferred site (Fig. 1d). The conversion rate for product **1a** rapidly increased within the first 2 min and then gradually decreased to 36.1%. Conversely, the accumulation of **1b** and **1c** gradually increased within 60 min. The total conversion rate of the three products reached a plateau at 30 min. These results indicate that the preferred catalytic site for AmAT7-3 is C3'-OH. Compound **1b** could either be a direct product of AmAT7-3, or an acyl-migration product of **1a** due to the adjacent hydroxyl groups on C3' and C4'[30]. To clarify this, **1a** was used as the substrate. When co-incubated with AmAT7-3, 41.8% of **1c** (C3',C4'-*O*-acetylated **1**) was generated. In contrast, when co-incubated with boiled AmAT7-3, **1c** was not detected (Fig. 1f). These results indicated that AmAT7-3 could modify the C4' site of **1a**. Additionally, the spontaneous conversion rate from **1a** to **1b** was around 6.1%, significantly lower than the **1b**/**1a** ratio of 56.5% in the catalytic product of AmAT7-3 (Fig. 1f vs 1c). This indicated that AmAT7-3 could also catalyze the C4' site of **1**. Based on the rapid generation of **1a**, we deduced that **1c** was produced by sequential acetylation at C3' and C4'.

### Proposed acetylation pathway of astragalosides in *A. membranaceus*

Besides isoastragaloside II (**1a**) and cyclocephaloside II (**1b**), saponins in *A. membranaceus* also include astragalosides II (**1d**) and I (**1e**), which are C2'- and C2', C3'-acetylated products of **1**, respectively (Fig. 2a). Since we screened 26 candidates and found no C2'-acetylation activity among them, we investigated the contribution of AmAT7-3 to the C2'-acetylated product. Firstly, the subcellular localization of AmAT7-3 was determined in *Nicotiana benthamiana* using the pSuper 1300-GFP vector (Fig. 2b). The fluorescent signal of GFP-AmAT7-3 was confined in the cytoplasm, indicating that AmAT7-3 is primarily expressed in the cytoplasm of living cells. In legume plants, the cytoplasmic environment tends to be slightly alkaline, as exemplified by *Vigna radiata* suspension-cultured cells (pH 7.5) and *Vicia faba* stomatal guard cells (pH 7.67)[31,32]. Thus, we investigated the stability of AmAT7-3 product **1a** under different pH conditions (Na$_2$HPO$_4$-NaH$_2$PO$_4$, pH 6.0, 7.0, 8.0) within 90 min (Fig. 2c). As a result, although acetylated saponins were prone to hydrolysis in alkaline environments (3.9%, 27.7%, 49.5% for pH 6, 7, 8, respectively), the proportion of C2'-acetylated product **1d** remarkably increased at pH 7 and pH 8. Among the non-hydrolyzed products, the proportion of **1d** was 4.8%, 34.2%, and 63.5% under pH 6, 7, and 8, respectively. To better understand the spontaneous acetyl migration reaction, the time course was obtained under 50 mM Na$_2$HPO$_4$-NaH$_2$PO$_4$ at pH 7.6, using **1a**/**1b**/**1d** (C3'/C4'/C2'-OAc) as the substrate. Interestingly, both **1a** and **1b** were convert to **1d** as a main acetylated product after 8 h (58.7–70.2% within the acetylated products), and only a small portion of **1d** was converted back to **1a**/**1b** (<25%) (Fig. 2d–f). These results indicated that **1a**/**1b**, the major products for AmAT7-3, could be converted into the dominant saponin **1d**

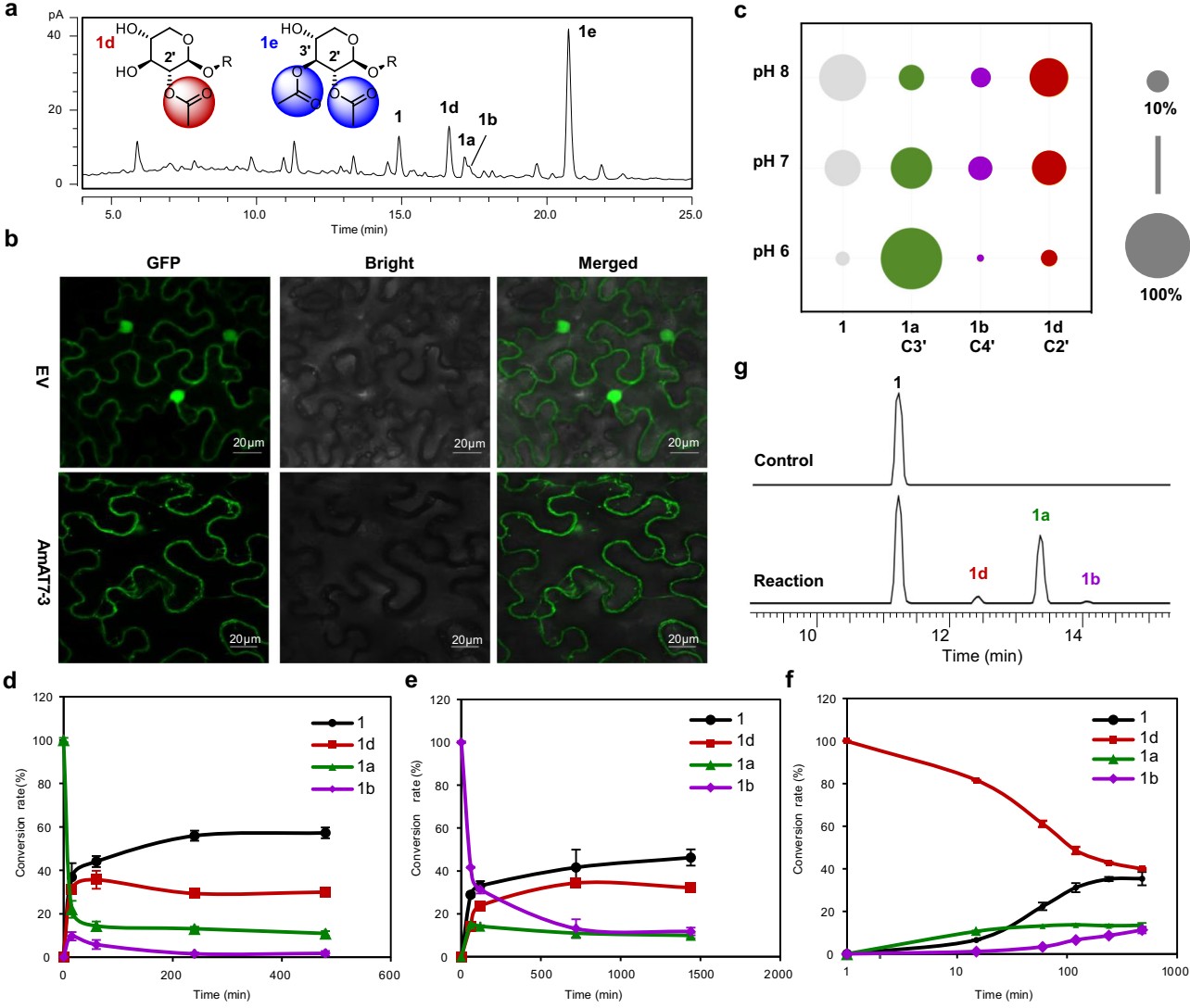

**Fig. 2 | Contribution of AmAT7-3 to C2′-OH acetylated saponins in *A. membranaceus*. a** UHPLC/CAD chromatogram of Astragalus Root. **b** Subcellular localization of AmAT7-3. Three independent experiments showed the similar results. **c** Proportions of degraded products of **1a** at different pH. Data are presented as the average of triplicate independent experiments. **d–f** Time-course of the acetyl migration for **1a**, **1b**, and **1d**, respectively. Data are presented in mean ± SD (*n* = 2 or

3 independent experiments). Source data are provided as a Source Data file for (**c–f**). **g** Transient expression of AmAT7-3 in *N. benthamiana*. UHPLC/MS traces for extracts from agroinfiltrated leaves are presented. EV: fluorescent signal of empty pSuper 1300-GFP vector; Reaction: transient expression AmAT7-3 in *N. benthamiana*; Control: transient expression empty pEAQ-*HT* vector in *N. benthamiana*.

in *A. membranaceus* through acetyl migration. The generation of **1d** could also be observed in *N. benthamiana* when AmAT7-3 was overexpressed and **1** was co-infiltrated as the substrate (Fig. 2g). This further suggests that acetylation products *in planta* are different from those in vitro, and AmAT7-3 may play a role in the production of the major astragalosides **1d** and **1e** through C3′/C4′-*O*-acetylation and acetyl-migration. The mechanism behind the accumulation of C2′-acetylated saponin in *A. membranaceus* requires further investigation.

### Crystal structure and site-specific catalytic mechanism of AmAT7-3

AmAT7-3 is a rare multi-site acetyltransferase which acetylate large saponin molecules. In order to gain insights into its acetylation mechanism, we solved the complex crystal structure of AmAT7-3/astragaloside IV (**1**) (PDB ID:8H8I) at a resolution of 2.03 Å. The crystal structure of saponin acyltransferases has not been reported previously, to the best of our knowledge. Attempts were also made to obtain the ternary complex crystal structure containing acetyl-CoA or CoA, but unfortunately not successful. Similar to previously reported

acyltransferases[17], the structure of AmAT7-3 was divided into two approximately equal-sized domains with a solvent channel running through the protein molecule (Fig. 3a). Notably, in comparison to the structure of SbHCT (PDB ID: 4KEC) which transfers *p*-coumaroyl-CoA to shikimic acid[16], the active pocket of AmAT7-3 was larger to accommodate saponin molecules (Fig. 3b–d).

To understand the mechanism of multisite acetylation, we conducted Molecular Dynamics (MD) simulations to investigate the binding and dynamics of substrate **1**. Since a reasonable position of Ac-CoA could not be obtained from docking (Supplementary Fig. 3), Ac-CoA was packed into the structure via superimposition based on the known structure of SbHCT[16]. The distance population between acyl *C* and various sugar *O* sites were shown in Fig. 3e. The shortest distances were observed for *Cx-O3′* (2.8 Å), *Cx-O4′* (3.1 Å), and *Cx-O2′* (3.8 Å), indicating that *O3′* and *O4′* were accessible for acetylation, while the *O2′* site may be less accessible. Due to the relatively spacious pocket of AmAT7-3, we observed significant rotational motion of the xylosyl sugar ring throughout the MD simulation. Based on the rotation angles, we segregated the substrate into two major conformations

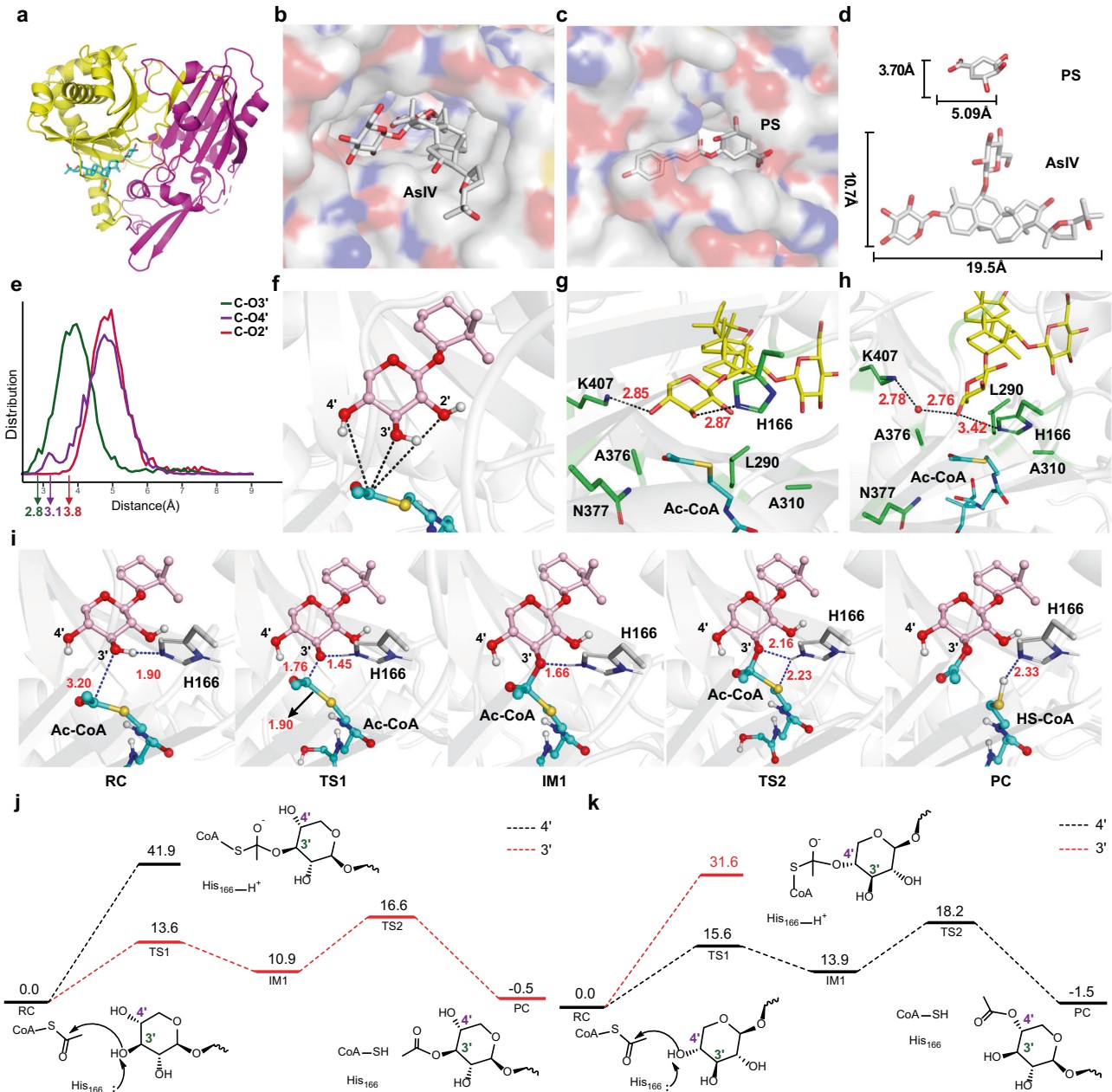

**Fig. 3 | Crystal structure and catalytic mechanism of AmAT7-3. a** The complex crystal structure of AmAT7-3/astragaloside IV. **b** The active pocket surface of AmAT7-3. **c** The active pocket surface of SbHCT. **d** Comparison of the molecular sizes of astragaloside IV and shikimic acid, substrates for AmAT7-3 and SbHCT, respectively. **e** The distance population between acyl *C* of Ac-CoA and sugar *O* sites of astragaloside IV. **f** The distance between acyl *C* and various sugar *O* sites of astragaloside IV. **g** A reactive snapshot of conformation-1 and its interaction with surrounding amino acids from MD simulations. **h** A reactive snapshot of conformation-2 and its interaction with surrounding amino acids from MD simulations. **i** QM/MM-optimized reaction states of the *O3′* acetylation reaction catalyzed by AmAT7-3. **j** QM/MM-calculated energy profile for AmAT7-3/ astragaloside IV/Ac-CoA at a reactive snapshot of conformation-1. **k** QM/MM-calculated energy profile for AmAT7-3/astragaloside IV/Ac-CoA at a reactive snapshot of conformation-2. AsIV astragaloside IV, PS *p*-coumaroyl shikimic acid, RC reactant complex, TS1 transition state 1, IM1 intermediate 1, TS2 transition state2, PC product complex. Energies are given in kcal/mol, while the distances are given in angstrom.

using a 75-degree threshold (Supplementary Fig. 4). In conformation-1, the *O3′* and *O4′* sites formed hydrogen bonds with K407 and H166, respectively. This H-bonding constrained the substrate, positioning the *O3′* site in close proximity to *C*x, implying a potential site for selective acetylation (Fig. 3g). As the sugar ring underwent rotation, it generated space for solvent water to enter, leading to the emergence of conformation-2. In this conformation, the *O4′* site approached *C*x, while the *O3′* site remained distant (Fig. 3h). Thus, the regioselectivity of acetylation could be logically explained by the substrate positioning in the two major conformations.

To further support the above predictions, combined quantum mechanics and molecular mechanics (QM/MM) computation was conducted on two reactive snapshots featuring close *C*x-*O3′* or *C*x-*O4′* distance from the near-attack conformation in either conformation-1 or conformation-2 (Supplementary Figs. 5–6). According to previous studies, the adjacent His residue likely acts as the acid-base catalyst during the acetylation reaction[16]. Our calculations showed that the reaction was initiated by the proton transfer from sugar *O* atom to His166, which was coupled to the nucleophilic attack of sugar *O* onto the acyl *C*x, leading to the formation of a tetrahedral intermediate.

Subsequently, another proton transfer occurred from His166 to $Sx$, facilitating the $C$-$S$ cleavage and resulting in the product **1a/1b** (Fig. 3i, Supplementary Fig. 7). The QM/MM-calculated energy profiles for two reactive snapshots of conformation-1 and conformation-2 are presented in Fig. 3j and Fig. 3k, respectively. Inspection of these energy profiles revealed a strong preference for acetylation at the $O3'$ site over the $O4'$ site in conformation-1, while the reverse was observed in conformation-2. The MD-predicted ratio of the two conformations was in qualitative agreement with experimental observations that the $O3'$ acetylation is favored over that of the $O4'$ site (Supplementary Fig. 4).

## Engineering the regioselectivity of AmAT7-3 and the stereo-selective mechanism of its mutants

To further elucidate how adjacent amino acids contributed to the conformation of the xylosyl group (Xyl) and regulated the regioselectivity of AmAT7-3, we analyzed the complex structure of AmAT7-3/astragaloside IV (**1**). In addition to the established critical role of H166 (Supplementary Fig. 8), ten amino acids within 5 Å from the substrate were found to participate in building the internal structure of the active cavity. These amino acids include K407, R38, D311, A312, A310, L290, I34, I288, I374, and A376 (Fig. 4a, Supplementary Data 2), which may play a decisive role in determining the substrate conformation. To investigate their roles, a mutant library was constructed, where seven amino acids were selected as substitutive residues, including Ala/Val (non-polar, aliphatic), Trp/Phe (non-polar, aromatic), Gly (polar, uncharged), Asp (acidic) and Arg (basic). The functions of the mutants were characterized using purified proteins following the same procedure as the wild-type. Astragaloside IV (**1**) and Ac-CoA were used as the substrate and acyl donor, respectively. Interestingly, several mutants remarkably enhanced the regioselectivity toward C3'-OH (Fig. 4b-d). For instance, the L290V mutant specifically modified C3'-OH with a higher conversion rate of 79.1%, compared to 35.1% of the wild-type. On the other hand, four mutants specifically modified the C4'-OH of **1**, as exemplified by A310W with a 53.1% conversion rate (19.8% for the wild-type). More interestingly, the mutants of A310 presented different selectivity (Fig. 4d). A310D/G/R selectively catalyzed the C3'-OH acetylation of **1**, while A310F/W specifically favored C4'-OH. In addition, we analyzed the kinetic parameters of the wild-type and the mutants (Supplementary Figs. 9-10). The $K_m$ and $k_{cat}/K_m$ value of AmAT7-3$_{A310G}$ toward **1a** were 76.7 μM and 0.6 mM$^{-1}$s$^{-1}$ while 168.3 μM and 0.19 mM$^{-1}$s$^{-1}$ for AmAT7-3$_{A310W}$ toward **1b**. Compared to the wild-type, the similarly-ranged kinetic parameters indicated that the mutations altered the regiospecificity without reducing the catalytic efficiency.

To elucidate the molecular determinants for the distinct regioselectivity, we solved the crystal structure of A310G in complex with substrate **1** (PDB: 8HBT, 1.96 Å). The structure of A310W variant was obtained from the wild-type AmAT7-3. Ac-CoA was docked into both structures in the same manner as with the wild-type protein. MD simulations showed that the substrate binding is relatively stable in both AmAT7-3$_{A310G}$ and AmAT7-3$_{A310W}$, with only one major conformation. To compare the position of the xylosyl group, the QM/MM-optimized structures of AmAT7-3$_{A310G}$ and AmAT7-3$_{A310W}$ were superimposed with the wild-type reactive snapshot of conformation-1, respectively (Fig. 4e, f). In the A310G variant, as the steric hindrance of the methyl group is removed from Ala, L290 was able to move closer to G310, leading to a simultaneous rotation of the sugar ring towards L290. On the other hand, the replacement of the Ala by the bulky Trp in A310W caused a significant steric effect, pushing L290 away from W310. The steric effect of L290 then pushed the sugar ring away. As a result, the sugar ring in AmAT7-3$_{A310G}$ rotated counterclockwise by 80° compared to that in the wild-type, while in A310W it rotated clockwise by 30°. Similar to A310W, steric effects were observed for A310F, A312W, and A312F variants, as demonstrated by MD simulations and mutagenesis results (Supplementary Fig. 11). Meanwhile, A310R and D311R could form hydrogen bonds with $O2'$, effectively stabilizing the

substrate and restricting the reaction to $O3'$ acetylation (Supplementary Fig. 11). The repositioning of the xylosyl group caused by these structural changes may be responsible for the observed regioselective acetylation in the respective variants.

Although the distance population between $Cx$-$O3'$ and $Cx$-$O4'$ in A310G/A310W was similar (2.8-3.3 Å, Supplementary Figs. 12-13), the vertical and spatial hindrance were significantly different. For A310G, $O3$ was almost perpendicular (92.9°) to the plane where the C = O group is located, while $O4$ formed an angle of 47.8° with C = O plane. The vertical position was more favorable for the nucleophilic attack reaction[33]. As a result, the $O3$- acetylation was kinetically favored over the $O4$- acetylation in AmAT7-3$_{A310G}$, which was consistent with the biochemical experimental results. QM/MM calculations showed that the energy barrier of the first reaction step was 19.9 kcal/mol for the $O3$-acetylation and 30.0 kcal/mol for the $O4$-acetylation, respectively (Fig. 4g). Regarding A310W, the presence of the Trp group induced steric hindrance not only to the sugar ring but also to Ac-CoA, impeding the acetylation of $O3$ (Supplementary Fig. 14). Consequently, the $O4$- acetylation is favored for AmAT7-3$_{A310W}$. QM/MM calculations predicted the energy barriers in A310W variant as 21.8 kcal/mol for $O4$-acetylation and 31.2 kcal/mol for $O3$-acetylation, respectively, which aligned with the observed regioselectivity of the variants. The computational results indicated that the regioselectivity was primarily decided by the positioning of sugar ring in the active pocket, which was controlled by the steric effects and hydrogen bonds from the surrounding residues.

To demonstrate the sequence and structure specificity for AmAT7-3, we aligned its protein sequence with six reported glycoside acetyltransferases (Fig. 4h, Supplementary Fig. 15). A conserved region YF(Y)GN was observed at #305-308, which was consistent with a previous study[34]. Adjacent to this conserved motif, a $\beta$10 barrel (#309-318) was located, which played a crucial role in forming the active pocket for AmAT7-3. Unlike other acetyltransferases, the amino acids composing the $\beta$10 barrel (AADAG) in AmAT7-3 were relatively small in size, resulting in a larger active pocket to accommodate the saponin molecules such as astragaloside IV. This was also consistent with the findings from the mutants A310, D311, and A312, where changes in catalytic activity were observed in most variants. These results indicated that the AADAG motif in the $\beta$10 barrel was crucial for the substrate recognition and the multi-site acetylation function of AmAT7-3. We also aligned the other residues forming the active pocket (#29-#34, #372-#377, #284-#291, #309-#318, #359-#365) with reported glycoside ATs (Supplementary Fig. 15). However, conserved or distinct motifs were not observed at these sites, despite most of the residues being aliphatic (L/A/P/I/V) at #289-290.

## Acetylation of medicinally important saponins using AmAT7-3 and its mutants

Acetylation has the potential to alter the activity and drugability of naturally occurring saponins[4-6]. However, chemical modification often encounters challenges of low selectivity due to the multiple hydroxyl groups on the sugar moiety. To address this issue, we utilized AmAT7-3 to acetylate eight medicinally important saponins, including astragaloside IV (**1**), astragaloside VII (**2**), ziyuglycoside II (**3**), bufalin 3-$O$-$\beta$-D-glucoside (**4**), cinobufagin 3-$O$-$\beta$-D-glucoside (**5**), 20($R$)-ginsenoside Rh1 (**6**), glycyrrhetinic acid 3-$O$-$\beta$-D-glucuronide (**7**), and chikusetsu saponin IVa (**8**). The bioactivities of these compounds were shown in Fig. 5a[35-41], where some of these original compounds or their derivatives, such as **1, 4, 5, 6**, and **7**, are currently undergoing clinical trials as drug candidates (NCT: 01553643, 00837239, 03843229, 00815763, and 05570110). The reaction mixtures were analyzed by UHPLC/MS, revealing that AmAT7-3 could effectively catalyze the regioselective acetylation of triterpenoid and steroidal saponins but not their aglycones (Supplementary Figs. 16-23). For example, AmAT7-3 catalyzed **6** with a conversion rate of 53.7%, versus 0% for its aglycone

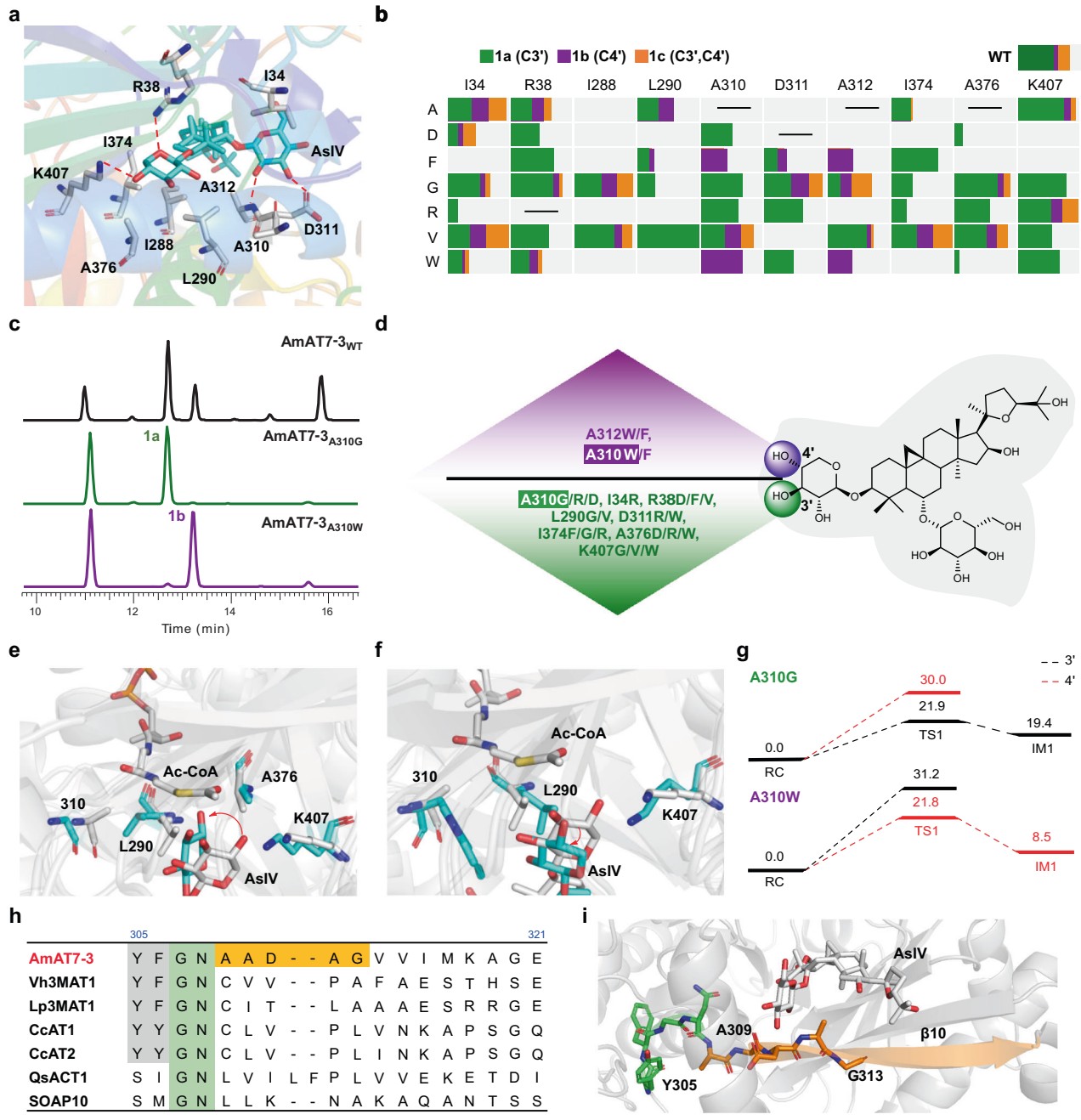

**Fig. 4 | Engineering the regioselectivity of AmAT7-3 and the regioselective mechanism of its mutants. a** Amino acids within 5 Å of **1** in the active pocket. **b** Mutant library of AmAT7-3 and the ratio of **1a/1b/1c** produced different mutants. **c** UHPLC/MS chromatograms of the reaction samples catalyzed by AmAT7-3 and its mutants. **d** Mutants with specific C3'-OH or C4'-OH acetylation activities. **e** Superposition between the QM/MM-optimized reactant complex of AmAT7-3$_{A310G}$ (cyan) and that of the wild-type (from conformation-1, gray). **f** Superposition

between the QM/MM-optimized reactant complex of AmAT7-3$_{A310W}$ (cyan) and that of the wild-type (from conformation-1, gray). **g** QM/MM-calculated energy profile (in kcal/mol) for AmAT7-3$_{A310G/W}$/**1**/Ac-CoA. AsIV, astragaloside IV (**1**). **h** Sequence alignment for AmAT7-3 and other flavonoid or saponin acetyltransferases. **i** Location of YFGN and AADAG motifs in the $\beta$10 barrel of AmAT7-3. RC reactant complex, TS1 transition state 1, IM1 intermediate 1.

protopanaxatriol (Fig. 5b). The enzyme exhibited remarkable selectivity to produce **6a**, a mono-acetylated saponin. Comparatively, in an optimized chemical acetylation reaction (acetic anhydride, 70 °C, 6 h, Supplementary Fig. 24)[42], at least 12 products were detected, including 3 mono-, 5 di- and 4 tri- acetylated compounds (Fig. 5b). These results highlighted the advantageous regioselectivity achieved through enzymatic acetylation using AmAT7-3.

In addition to the xylosyl group (**1-2**), AmAT7-3 could also recognize other sugar moieties including glucosyl (**4-6**), arabinosyl (**3**) and glucuronic acid (**7-8**) (Fig. 5c). Four mono-acetylated products

were prepared, and their structures were characterized using NMR spectroscopic analyses (Supplementary Figs. 25–53, Supplementary Tables 3–6). Among them, **3a**, **3b** and **7a** were not reported previously. The NMR spectra revealed exhibited new carbon signals ($\delta_C$ 171.0–171.3) and new methyl signals ($\delta_H$ 1.97–2.11, s, 3H) in all four products, indicating successful acetylation. The site of acetylation was further confirmed through HMBC correlation. For example, in product **3a**, the HMBC correlations were observed between H-3 ($\delta_H$ 3.33) and C-2 ($\delta_C$ 27.0)/C-4 ($\delta_C$ 39.9), as well as with the newly appeared carbonyl carbon ($\delta_C$ 171.3), allowing the identification of compound **3a**

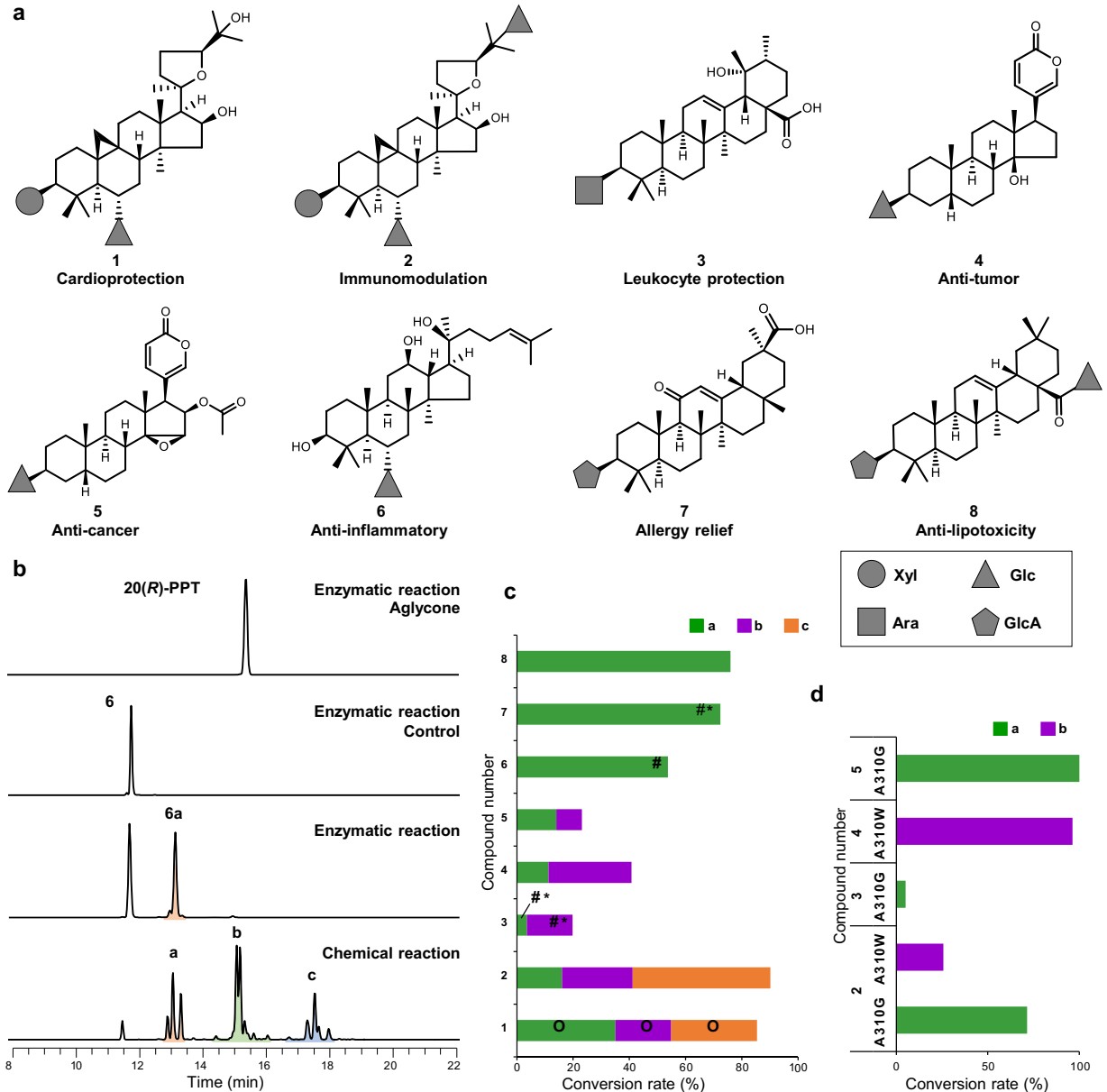

**Fig. 5 | Acetylation of medicinally important saponins using AmAT7-3 and its mutants. a** Structures and bioactivities of compounds **1-8**. **b** Acetylated reaction of **6** catalyzed by AmAT7-3 or by chemical reagents. PPT protopanaxatriol, a/b/c indicated mono-/di-/try-acetylated products. **c** Conversion rates of acetylated products catalyzed by AmAT7-3. O: identified by comparing with the standards; #: prepared and identified by NMR; *: compounds which were not reported previously. **d** Conversion rates of acetylated products catalyzed by AmAT7-3$_{A310G}$ and AmAT7-3$_{A310W}$.

as C3′-O-acetate ziyuglycoside II. Similarly, products **6a, 7a**, and **3b** was identified as the acetylation products of C6′-OH, C3′-OH, and C4′-OH, respectively. These results demonstrate the versatility of AmAT7-3 in reacting with various sugar moieties and different sites on the saponin molecules. Based on the examination of ligand interaction patterns (Supplementary Fig. 54), the structural basis behind substrate promiscuity likely stems from the large volume of the active pocket, facilitating the accommodation of sizable saponin molecules. The positioning of the saponins could be determined by conserved amino acids such as R38 and E39, which establish hydrogen bonds with the sugar moiety.

Though single products were generated for compounds **6-8**, AmAT7-3 catalyzed compounds **2-5** to form 2 or 3 products, similar to the reaction observed with compound **1**. To improve the

regioselectivity, we utilized the regio-selective mutants, AmAT7-3$_{A310G}$ and AmAT7-3$_{A310W}$ to catalyze the acetylation for **2-5** (Fig. 5d). As expected, the mutant showed higher regioselectivity compared to the wild-type enzyme. For instance, the conversion rate of the wile-type enzyme for **5** was 23.1%, containing two products in the ratio of 14:9. The A310G variant catalyzed **5** to generate a single product with a conversion rate of 96.3%. These findings further confirm the crucial role of residue #310 in shaping the active cavity and highlight the potential of regiospecific acetylation for preparing mono-acetylated products in the future.

In summary, we characterized a saponin acetyltransferase AmAT7-3 from the medicinal plant *Astragalus membranaceus* using a stepwise BLAST approach. We solved the crystal structures of AmAT7-3 and its variant A310G in complex with astragaloside IV, respectively.

AmAT7-3 exhibits a distinct protein sequence compared to reported acetyltransferases, and represents the rare saponin acetyltransferase with a crystal structure. It could catalyze C3′/C4′-*O*-acetylation of astragaloside IV (**1**) and the acetylation of other seven medicinally important saponins. A total of four products were prepared and three of them were not reported previously, to the best of our knowledge. Structural analysis revealed that AmAT7-3 possesses a large active pocket decided by a specific sequence AADAG, and its regioselectivity is controlled by the positioning of the sugar ring within the active pocket through key amino acids such as A310, L290, K407, D311, and A312. To enhance its regiospecificity, a small mutant library was constructed based on semi-rational design. The variants A310G and A310W were found to catalyze specific C3′-*O* and C4′-*O* acetylation for astragaloside IV (**1**), respectively. Moreover, these mutants demonstrated higher regiospecificity when modifying bioactive saponins **2-5**. This work provides insights into the protein sequences, catalytic mechanisms and semi-rational engineering of glycoside acetyltransferases. It also presents efficient catalytic tools for saponin acetylation.

## Methods

### General methods

Compounds astragaloside IV (**1**), ziyuglycoside II (**3**), 20(*R*)-ginsenoside Rh1 (**6**), glycyrrhetinic acid 3-*O*-β-D-glucuronide (**7**), chikusetsusaponin IVa (**8**), isoastragaloside II (**1a**), cyclocephaloside II (**1b**), 3-*O*-3′,4′-diacetyl-β-D-xylopyranosyl-6-*O*-β-D-glucopyranosyl-cycloastragenol (**1c**), astragaloside II (**1d**), and astragaloside I (**1e**) were purchased from Must Bio-Technology (Chengdu, China). Astragaloside VII (**2**), bufalin 3-*O*-β-D-glucoside (**4**) and cinobufagin 3-*O*-β-D-glucoside (**5**) were prepared as previously reported[27,37,43]. Acetyl-CoA was purchased from ZZBIO (Shanghai, China). Methanol and acetonitrile (UPLC grade) were purchased from Fisher Scientific (USA).

### Plant materials

Seeds of *A. membranaceus* were purchased from Anguo FengHua Seed Station (Hebei, China). Two-week-old seedlings were obtained following previous published methods[27] and used for gene cloning. An Astragalus Root sample was collected from Daxing'anling (Heilongjiang, China) and used for chemical analysis.

### Phylogenetic analysis

The evolutionary history was inferred using the Maximum Likelihood method based on the JTT matrix-based model[44], and evolutionary analyses were conducted in MEGA7[45] in this study. Initial tree(s) for the heuristic search were obtained automatically by applying Neighbor-Join and BioNJ algorithms to a matrix of pairwise distances estimated using a JTT model. The topology with the superior log-likelihood value was selected. The bootstrap value was 1000.

### Gene cloning and protein expression

Total RNA was extracted using the TranZol™ kit (Transgen Biotech, China) and reverse-transcribed to cDNA using FastQuant RT Kit (Tiangen Biotech, China). Specific primers were designed to amplify *AmAT7-3* and *AmAT8-17*, which was subsequently cloned into the pET-28a(+) vector (Invitrogen, USA). The mutant plasmids were constructed by the specific primers in Supplementary Data 2 using the Fastpfu Fly colony kit (TransGen Biotech, China). After sequencing, the recombined plasmid was transformed into *E. coli* BL21 (DE3) (TransGen Biotech, China). The bacterial strain was cultured in Luria-Bertani (LB) medium supplemented with 50 μg/mL kanamycin at 37 °C until the OD600 reached 0.6. Isopropyl β-D-thiogalactoside (IPTG, 0.1 mM) was then added to induce protein expression, and the culture was further incubated for 20 h at 16 °C.

To purify the protein, the bacterial cells were harvested by centrifugation (6010 *g*, 3 min, 4 °C) and resuspended in 15 mL of lysis buffer (pH 8.0, 50 mM NaH₂PO₄, 30 mM NaCl, 10 mM imidazole). The cells were then disrupted by ultrasonication on ice for 10 min, and the lysate was clarified by centrifugation at 13523 *g* for 45 min to remove the debris. The supernatant was purified using a Ni-NTA column (Proteinlso Ni-NTA Resin, TransGen Biotech, Beijing, China) and the protein was concentrated using Amicon Ultra-15 Ultracel-30K centrifuge filters (Merck Millipore).

### Protein purification and crystallization

To perform kinetic analysis and crystallization, AmAT7-3 was further cloned into pET28a-TEV vector through specific primers and purified by size exclusion chromatography on a Superdex™ column (GE healthcare) (Supplementary Table 2). The protein concentration was determined using the Protein Quantitative Kit (TransGen Biotech, China) and the purity was assessed by SDS-PAGE as shown in Supplementary Fig. 4.

Purified protein (20 mg/mL for AmAT7-3 and AmAT7-3₍A310G₎) was incubated with 5 mM Ac-CoA and 5 mM astragaloside IV (**1**) on ice for 1 h. The crystals of AmAT7-3/**1** and AmAT7-3₍A310G₎/**1** were prepared by hanging drop vapor diffusion. Cubic crystals were obtained in hanging drops containing 1 μL of protein solution and 1 μL of reservoir solution (1.6 M magnesium sulfate, 100 mM MES/sodium hydroxide pH 6.5) after 3 weeks. Crystal diffraction and data collection were carried out on beamline BL19U1 at the Shanghai Synchrotron Radiation Facility using HKL-2000 program suite. Initial crystallographic phases were constructed through molecular replacement based on the predicted structure of AmAT7-3 by alphafold 2. Protein models were built and refined in CCP4, Wincoot and Phenix, respectively. Detailed information of the structures is listed in Supplementary Table 7. The *Fo-Fc* omit map for ligand in the crystal structures were shown using Pymol (Supplementary Fig. 55).

### UHPLC/MS analysis of enzymatic reaction products

To characterize the catalytic activity, a 100-μL reaction mixture was prepared by mixing 30 μg purified protein, 0.1 mM astragaloside IV (**1**), 0.5 mM Ac-CoA and 0.5 mM dithiothreitol in 50 mM Na₂HPO₄-NaH₂PO₄ buffer (pH 6.0) at 30 °C for 30 min. The reaction mixture was quenched with 200 μL of methanol, concentrated to dryness, reconstituted with 200 μL of methanol, and centrifuged at 21130 *g* for 20 min. The resulting supernatant was subjected to UHPLC/MS detection using a Vanquish UHPLC system coupled with a Q-Exactive quadrupole-orbitrap mass spectrometer equipped with a heated electrospray ionization source (ThermoFisher Scientific, USA). The samples were separated on a Waters T3 column (2.1 × 100 mm, 1.8 μm) using 0.1% formic acid (*v*/*v*, A) and acetonitrile (B) at 50 °C. A gradient program was used: 0–1 min, 8% B; 1–19 min, 8–47% B; 10–16 min, 47–65% B; 16–19 min, 65–100% B; 19–22 min, 100% B. The flow rate was 0.3 mL/min. MS analysis was performed in both positive and negative ion modes with the following parameters: spray voltage: ±3.5 kV; capillary temperature: 350 °C; sheath gas: 45 arb; aux gas: 10 arb; probe heater temperature: 400 °C; S-lens RF level: 60 V; resolution: 70000 for full MS and 17500 for MS/MS; scan range: *m/z* 100–1200; stepped NCE: 35 eV. Data were processed using Xcalibur™ 4.1 software (Thermo Fisher Scientific). The reaction mixture was separated by HPLC (SSI-III) coupled with ELSD 6000 (Alletch Chrom, USA).

### Transient expression in *N. benthamiana*

AmAT7-3 was initially cloned into the pDnor207 vector using specific primers and then introduced into pEAQ-*HT* vector by Gateway Reaction (ThermoFisher Scientific, USA) (Supplementary Table 2). The recombinant plasmid was transferred into *Agrobacterium tumefaciens* strain LBA4404. The strain was cultured at 28 °C until the OD600 reached 1.0. The bacterial culture was centrifuged at 1697 *g* for 10 min and re-suspended in MMA solution (10 mM MES, 10 mM MgCl₂, 150 μM acetosyringone, pH 5.6) to an OD600 of 0.2. Then it was incubated in the dark at 28 °C for an hour and infiltrated into the leaves of 4-week-

old tobacco plants. Three days later, 0.1 mM astragaloside IV was injected into the leaves and the plants were cultivated for another 4 days. The leaves were cut into small pieces for chemical analysis using the same method mentioned above[4].

## System setup for computational studies

The initial structure of the enzyme was obtained from the crystal structure of AmAT7-3 protein (PDB: 8H8I, 2.03 Å). The protonation states of titratable residues (His, Glu, Asp) was assigned based on PROPKA[46] values and local hydrogen-bonded network. The coordinates of Ac-CoA were obtained from the superposition between 8H8I and 4KEC (PDB: 4KEC). AmAT7-3$_{A310G}$ (PDB: 8HBT, 1.96 Å) was processed with the same method. The mutant AmAT7-3$_{A310W}$ was obtained from the structures of wild-type (AmAT7-3, astragaloside IV, Ac-CoA) by using mutation tools in pymol. Protein residues and ligands were treated by Amber ff14SB force field and general AMBER force field (GAFF), respectively[47]. Partial atomic charges of substrates and acetyl-CoA were calculated using the RESP approach at the B3LYP/def2-TZVP theory level. Subsequently, sodium ions were added to the surface of the protein to balance the overall charge of the complex systems. The entire system was solvated in a rectangular box with TIP3P waters (minimum distance was 15 Å from the protein surface).

## Classical MD simulations

The complex system first was subjected to 5000 steps of steepest descent and 5000 steps of conjugate gradient minimization, with the protein held fixed by using position restraints with a force constant of 500 kcal·mol−1 Å−2. And an additional 5000 steps of steepest descent and 25,000 steps of conjugate gradient minimization was performed to fully optimize the system without restraints. Then, the systems were annealed from 0 to 300 K for 50 ps with the NVT ensemble, during which the constraint of 15 kcal/mol/Å was applied. Then, the density equilibrium for 1.0 ns was conducted under the NPT ensemble to obtain a uniform density, where the target temperature of 300 K was kept with the Langevin thermostat[48] and a 2 ps collision frequency, and the 1.0 atm target pressure was maintained with the Berendsen barostat[49] and a pressure relaxation time of 1 ps. Subsequently, all the restraints on the complex systems were removed, and the enzyme complexes were equilibrated for 4 ns under the NPT ensemble. Lastly, a productive MD simulation of 200 ns was carried out under the NPT ensemble. During the simulations, the covalent bonds involving hydrogen atoms were constrained with the SHAKE method[50], and the integration step was set to 2 fs. A cutoff radius of 8 Å was set for nonbonded interactions, while the long-range electrostatic interactions were treated using the Particle Mesh Ewald (PME) method[51]. All MD simulations were performed using Amber 18 software package[52].

## QM/MM calculation

Based on the MD simulation of wild-type protein and mutants, major conformations were observed in simulation trajectory. Reactive snapshots with close distances between acyl *C* and sugar *O* sites in the major conformation were selected to facilitate the QM/MM calculations. The QM/MM calculations were conducted using ChemShell, which uses turbomole[53] for QM region and DL_POLY[54] for MM region. Electronic embedding scheme[55] was employed to account the polarizing effect of the enzyme environment in the QM region, while the hydrogen link atoms with the charge-shift model was used to deal with the QM/MM boundary. QM region consisted of astragaloside IV, Ac-CoA and the surrounding key residues. It was treated by the hybrid B3LYP functional[56] with the all-electron basis set of def2-SVP, whereas the remaining MM part of the system was modeled at the classical level using the same parameters as in the classical MD simulations. Energies were further corrected with the large all-electron basis set def2-TZVP Dispersion energies were included in both optimizations and single-point energy calculations[57,58]. The limited memory quasi-Newton

(L-BFGS) algorithm in the DL-FIND optimizer was used in the QM/MM geometry optimization in Chemshell[59–62], in which the default convergence criteria have been used. The transition states were determined as the highest energy structure from the adiabatic scan, which was further optimized using the P-RFO optimizer[63] without any restraints. According to our previous and other groups' work, entropy correction usually plays minor effects in the catalytic reactions of enzymes[64–66]. Given the high computational cost for the QM/MM free energy calculations, we use the electronic energy barriers as estimates of the free energy barriers in the enzyme, which was demonstrated to be practical in previous work[67–69].

## Reporting summary

Further information on research design is available in the Nature Portfolio Reporting Summary linked to this article.

## Data availability

The sequences mentioned in this study have been deposited in NCBI, the accession number of AmAT7-3 is ON804888; the accession number of AmAT8-17 is OQ915518. The crystal structures obtained in this study have been deposited in the Protein Data Bank: AmAT7-3/astragaloside IV PDB 8H8I, AmAT7-3$_{A310G}$/astragaloside IV PDB 8HBT. Source data are provided with this paper, and data are available on request.

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

## Acknowledgements

This work was supported by National Natural Science Foundation of China (82122073, X.Q.; 81973448, X.Q.). We gratefully acknowledge Dr. Fuling Yin and Hongli Jia at State Key Laboratory of Natural and Biomimetic Drugs of Peking University for technical help in the crystallization experiments. Special thanks to Ying Fu (College of Biological Sciences, China Agricultural University) for the assistance of subcellular localization plasmid pSuper 1300-GFP. We also extend our gratitude to Dr. Huan Zhou (Shanghai Synchrotron Radiation Facility, Shanghai Advanced Research Institute, Chinese Academy of Sciences) for the crystal diffraction and data collection. We gratefully acknowledge Baiying Xing (Peking University) for her help in the crystal structure elucidation. We thank Bin Li (Peking Union Medical College) and Pi Li (ThermoFisher Scientific) for valuable technical consultation. We are grateful to Prof. George Lomonossoff at John Innes Centre for providing the pEAQ-*HT* vector.

## Author contributions

L.W., J.Z., and K.C. conducted the characterization, crystallization, and mutagenesis experiments. Z.J. and B.W. performed the theoretical calculations. M.Z. and Z.W. performed the X-ray crystallography and analyzed the crystal structure. M.Y. and X.Q. designed the experiments. L.W., Z.J., B.W. and X.Q. analyzed the data and wrote the paper.

## Competing interests

The authors declare no competing interests.
