## [Peer Review File · Nature Communications]

Reviewers' Comments:

Reviewer #1:

Remarks to the Author:

Qiao et al. reported a unique saponin acetyltransferase AmAT7-3 from *Astragalus membranaceus*, which catalyzing the regiospecific acetylation on saponin C-3 sugar moiety. They also resolved its crystal structure and correspondingly variant A310G to discuss the possible catalytic mechanism. This work expands the understanding of saponin acetyltransferases, and could also provide efficient catalytic tools for saponin acetylation. However, some issues need to be replied before it could be published in Nature communications.

1. title page: the authors address is "Linlin Wang^{1†}, Zhihui Jiang^{3†}, Jiahe Zhang¹, Kuan Chen¹, Meng Zhang¹, Zilong Wang, Binju Wang^{3*}, Min Ye^{1,2*} and Xue Qiao^{1,2*}", well, is there some mistakes? Or should be "Linlin Wang^{1†}, Zhihui Jiang^{2†}, Jiahe Zhang¹, Kuan Chen¹, Meng Zhang¹, Zilong Wang, Binju Wang^{2*}, Min Ye^{1,3*} and Xue Qiao^{1,3*}"?

2. Introduction: Sentences 1 to 2, it is not smooth in logical. All the introduction part is focus on Saponins, but what is the meaning of "In living organisms, acetylation is one of the most common modifications types and played an important role in efficacy or bioactivity for saponins", especially "In living organisms". In addition, "efficacy" and "bioactivity" seems have a same meaning. Maybe, you could readjustment the sequences of the sentences and follow the logical below: "In living organisms, acetylation is one of the most common modifications types and played an important role in improving the bioactivity of natural products. One example is Saponins..... For example,....."

Results and Discussion:

3. Line 118: *E. coli* should *Escherichia coli* when it first appeared.

4. Line 120 The second "purified" could be deleted.

Line 121 (pH 6.0). The reaction was conducted in 30 °C for 30 min. Or, the specific method can be supplied in SI.

5. 129: Should be "Fig. 1A and 1B".

The detailed experimental should be addressed in Figure S1, 2, 4.

How do you count the conversion rate?

6. 178 The title is Biosynthesis of astragalosides in *A. membranaceus*, in fact, you only using *planta N. benthamiana* to test its activity and it should be changed. Or, you may want to investigate the biosynthesis mechanism of astragalosides in *A. membranaceus*, from this point of view, it is certainly OK, while, there may ambiguity.

7. Line 241: acetylation reaction.16 should be changed into acetylation reaction 16.

MD and QM/MM paly a main role in mechanism investigating, while it is absent in the abstract parts. And their full names are also needed.

8. Line 248: in line with our MD prediction, what is the MD prediction? Where is it?

Reviewer #2:

Remarks to the Author:

The manuscript "Characterization, structural basis and protein engineering of a regiospecific saponin acetyltransferase from *Astragalus membranaceus*" by Wang et al. analyses the catalytic mechanism of acetylation, possible reasons for regio-selectivity in the active-site structure and substrate position and uses this information to engineer mutants with increased regio-selectivity on saponon variants. Crystal structures and mutation experiments are accompanied by modelling studies.

This work contributes to our understanding of natural glycoside acylation and gives examples of engineering enzymatic regio-selectivity by single-site mutations and is therefore in principal suitable for Nature communications.

There are, however, a number of points that must be addressed.

Since I am no expert in the experimental part, this review focusses on the computational studies

and some general remarks/questions.

The authors rationalise regio-selectivity by substrate positioning. However, this is by their construction, i.e. placement of the Ac-CoA. What does the complex look like, and how do the relevant positions behave, with the substrate positioned in a way that would contradict the observed regio-selectivity? That can be answered by modelling a negative control structure. Or there is simply no space for such a structure, which could also be mentioned/shown.

Although packing substrates in analogy to their binding site in related complexes is often successful and the proposed structures look reasonable, the authors want to check this. A quick test is to also blindly dock Ac-CoA into the crystal structure of the protein and check if the results agree with the assumed binding site and posed. Ideally, the same is done with the known complex structure to validate the docking approach.

The authors report a water molecule entering between K407 and the substrate in the course of the simulation. Is this a bulk water molecule or a crystal water molecule (if there were any crystal water molecules to start with)?

Should the two conformation, with different Cx-O3' distances, not show as a bimodal distribution? Figure 3E shows a small shoulder in the Cx-O4' distance distribution but nothing further that indicates two major conformations.

Have the two conformations been observed interchanging between each other? Or, as the text suggest, was one the starting conformation and the other one has emerged in the course of the MD simulations? In the latter case, the second conformation, the one appearing later in the simulation, is more likely to be a relaxed, equilibrated structure. Whether this is indeed the case (and with 50ns there is a chance it is not) cannot be told. The authors should therefore provide the time series of important distances in the supplementary material.

H166 plays an important role in the suggested mechanism by accepting a proton from the substrate. Has the possibility of a protonated H166 been evaluated? PROPKA, which was used to find the initial protonation states of titratable residues is limited when small-molecule ligands in the vicinity have to be taken into account. Has the substrate been considered when assigning protonation states? It would also be an easy control to run a short (i.e. another 50ns) MD simulation with protonated H166 to check (and perhaps rule out) this possibility.

Among the many mutants there is no H166X mutation. Would that not be a good control of the importance of H166 (if not of its exact role)?

A310F shows a preference for C4' acetylation, similar to the A310W mutant. This is quite likely for sterical reasons as discussed for A310W.

Similar effects are also observed when mutating A312 which has a different location and might "push" in another direction. It would be worth making a model of those mutants that show the sterical demands (no reaction calculation required).

All the other mutations that have an effect on activity, rather than on regio-selectivity are very interesting. For example, mutating A310 to R or D does not make much of a difference. D311 can be mutated to R without apparent consequences, although one would assume the hydrogen-bonding pattern, and thus the binding mode of the substrate, to change significantly. Without further (modelled) structures, this will be a bit speculative but perhaps the authors can discuss more of the mutation effects on activity based on their reaction calculations.

The classical MD simulations, performed prior to the QM/MM calculations already suggest a conformational flexibility of the substrate. The authors tried to take this into account by modelling the reaction starting from two different conformations. Their optimisation-only approach, however, does not allow conformational changes in the course of the reaction or does it account for substrate or protein flexibility and possible entropic effects associated therewith. This should at least be discussed (since QM/MM free energy calculations are far more demanding than the optimisations used in the present work and more modelling/simulations are needed for testing binding poses, protonation states and consequences for the mechanism already).

There are also some details missing in the methods section of the computational studies:

How many steps, using which optimisation algorithm were used for minimisation, equilibration and annealing? Which thermostat was employed and which barostat (if applicable) to control which temperature (and pressure)?

What are "major conformations observed in the trajectory"? How were "representative snapshots" chosen, i.e. in which respect are they representative?

Please provide the exact composition of the QM region, i.e. list all residues and positions of link atom placement.

Which optimisation algorithm was used and what were the convergence criteria? How were transition states searched (i.e. guessing transition structures, potential energy scan, chain-of-state optimisation) and which algorithm was used for that?

Reviewer #3:

Remarks to the Author:

In this manuscript, the authors report the structure and function analysis of saponin acetyltransferase from *Astragalus membranaceus*. The acetyltransferase AmAT7-3 which catalyzes acetylation of C3'/C4' position of astragaloside IV was identified by genome mining method. The crystal structure, MD simulation, and QM/MM analysis of AmAT7-3 revealed the key active site residues for regioselectivity and the two major conformations of substrate in the active site. Furthermore, the authors successfully altered the regioselectivity by mutagenesis experiments. The authors also demonstrated the generation of acetylated compounds by using various substrate analogs. This manuscript is well-written and easy to follow. The detailed structural and calculation analysis of the enzyme provide detailed information on regioselectivity of acetylation reaction. The results shown in this manuscript would attract the interests of the researchers in the field of natural product chemistry, biosynthesis, and enzymology. Therefore, I would recommend it for acceptance with some revisions.

1. To further analysis of the acetyl migration, time course reaction should be performed.
2. The authors should comment on the relationship between the ratio of conformation-1/conformation-2 and the ratio of O3'-/O4'-products.
3. The sequence comparison among AmAT7-3 and homologues around L290 region should also be shown.
4. The key HMBC correlations between acetyl group and sugar are not clear in some NMR figures. The closed-up views should be added.
5. Interactions between aglycon and active site residues should be more discussed. In addition, the authors should explain the structural basis for the broad substrate specificity of AmAT7-3 toward various aglycons.
6. The Fo-Fc omit map for ligand in the crystal structure need to be added in SI.

Response to reviewers' comments

Responses to Reviewers' comments

Reviewer #1

Qiao et al. reported a unique saponin acetyltransferase AmAT7-3 from *Astragalus membranaceus*, which catalyzing the regiospecific acetylation on saponin C-3 sugar moiety. They also resolved its crystal structure and correspondingly variant A310G to discuss the possible catalytic mechanism. This work expands the understanding of saponin acetyltransferases, and could also provide efficient catalytic tools for saponin acetylation. However, some issues need to be replied before it could be published in *Nature communications*.

1. title page: the authors address is “Linlin Wang^{1†}, Zhihui Jiang^{3†}, Jiahe Zhang¹, Kuan Chen¹, Meng Zhang¹, Zilong Wang, Binju Wang^{3*}, Min Ye^{1,2*} and Xue Qiao^{1,2*}”, well, is there some mistakes? Or should be “Linlin Wang^{1†}, Zhihui Jiang^{2†}, Jiahe Zhang¹, Kuan Chen¹, Meng Zhang¹, Zilong Wang, Binju Wang^{2*}, Min Ye^{1,3*} and Xue Qiao^{1,3*}”?

R: We thank the reviewer for reviewing our manuscript carefully. We have listed addresses in the same order as the author list. It is modified as follows:

Linlin Wang^{1†}, Zhihui Jiang^{2†}, Jiahe Zhang^{1†}, Kuan Chen¹, Meng Zhang¹, Zilong Wang¹, Binju Wang^{2*}, Min Ye^{1,3*} and Xue Qiao^{1,3*}

¹ State Key Laboratory of Natural and Biomimetic Drugs, School of Pharmaceutical Sciences, Peking University, 38 Xueyuan Road, Beijing 100191, China

² State Key Laboratory of Physical Chemistry of Solid Surfaces and Fujian Provincial Key Laboratory of Theoretical and Computational Chemistry, College of Chemistry and Chemical Engineering, Xiamen University, 361005 Xiamen, China

³ Peking University-Yunnan Baiyao International Medical Research Center, 38 Xueyuan Road, Beijing 100191, China

2. Introduction: Sentences 1 to 2, it is not smooth in logical. All the introduction part is focus on Saponins, but what is the meaning of “In living organisms, acetylation is one of the most common modifications types and played an important role in efficacy or bioactivity for saponins”, especially “In living organisms”. In addition, “efficacy” and

“bioactivity” seems have a same meaning. Maybe, you could readjustment the sequences of the sentences and follow the logical below: “In living organisms, acetylation is one of the most common modifications types and played an important role in improving the bioactivity of natural products. One example is Saponins..... For example,

R: We thank the reviewer for the valuable comments. This sentence was revised as: “Saponins are widely distributed and structurally complex natural products with various bioactivities. Acetylation is one of the most common modifications for saponins, and plays an important role in improving their bioactivity”.

3. Line 118: E. coli should Escherichia coli when it first appeared.

R: We have added the full name and defined the abbreviations was *E. coli* here. The revised sentence is: “Subsequently, these candidate genes were cloned into pET-28a (+) vectors, and the proteins were expressed in *Escherichia coli* (*E. coli*).”

4. Line 120 The second “purified” could be deleted. Line 121 (pH 6.0). The reaction was conducted in 30 °C for 30 min. Or, the specific method can be supplied in SI.

R: We thank the reviewer for the comment. We have carefully revised our description regarding protein functional characterization. The revised sentences are: “The function of the purified recombinant proteins was characterized in a mixture including 30 µg protein, 0.1 mM astragaloside IV (**1**), 0.5 mM acetyl-CoA and 0.5 mM dithiothreitol in 50 mM Na₂HPO₄-NaH₂PO₄ buffer (pH 6.0, 100 µL). The reaction was conducted in 30 °C for 30 min.” The reaction condition was included here to facilitate the comparison with the stability tests of the products acetylated products under different pH values (Figure 2C-F).

5. Line 129: Should be “Fig. 1A and 1B”. The detailed experimental should be addressed in Figure S1, 2, 4. How do you count the conversion rate?

R: We have cited Fig. 1A and 1B when discussing the phylogenetic analysis. Experimental details were added for Figures S1, S2 and S4, respectively.

For Supplementary Fig. 1: The function of the recombinant protein was characterized in a mixture including 30 µg protein, 0.1 mM astragaloside IV (**1**), 0.5 mM acetyl-CoA and 0.5 mM dithiothreitol in 50 mM Na₂HPO₄-NaH₂PO₄ buffer (pH 6.0, 100 µL). The reaction was conducted in 30 °C for 30 min. The samples were analyzed following the “UHPLC/MS analysis of enzymatic reaction products” section in Materials and Methods.

For Supplementary Fig. 2: Astragaloside IV (**1**) was used as acyl acceptor and Ac-CoA as acetyl donor. The optimized reaction condition was at pH 6.0 (50 mM NaH₂PO₄-Na₂HPO₄) and incubated at 30°C for 30 min. Unless otherwise specified, all parameters remain consistent with the optimal conditions in each panel. NBS: citric acid-sodium citrate buffer (pH 4.0-6.0); PBS: Na₂HPO₄-NaH₂PO₄ buffer (pH 6.0-8.0); Tris: Tris-HCl buffer (pH 7.0-9.0); CBS: Na₂CO₃-NaHCO₃ buffer (pH 9.0-11.0).

For Supplementary Fig. 10 (previous Fig. S4): The apparent K_m value was determined using isoastragaloside II (**1a**, iAsII) or cyclocephaloside II (**1b**, CycII) as the acyl acceptor and Ac-CoA as the acyl donor at 30 °C for 10 min in Na₂HPO₄-NaH₂PO₄ (pH 6.0). The samples were analyzed following the “UHPLC/MS analysis of enzymatic reaction products” section in Materials and Methods.

The relative conversion rate was calculated as follows and has been added to the legend of Supplementary Fig. 2:

$$\text{Conversion rate} = A_p / (A_p + A_s) * 100\%$$

$$\text{Relative conversion rate} = C_c / C_m * 100\%$$

A_p : peak area of the product; A_s : peak area of the substrate; C_c : conversion rate under the described condition; C_m : the highest conversion rate among all conditions.

6. Line 178: The title is Biosynthesis of astragalosides in *A. membranaceus*, in fact, you only using planta *N. benthamiana* to test its activity and it should be changed. Or, you may want to investigate the biosynthesis mechanism of astragalosides in *A. membranaceus*, from this point of view, it is certainly OK, while, there may ambiguity.

R: We thank the reviewer for the suggestion. We have changed the subtitle into “Proposed acetylation pathway of astragalosides in *A. membranaceus*”.

7. Line 241: acetylation reaction.¹⁶ should be changed into acetylation reaction ¹⁶. MD and QM/MM play a main role in mechanism investigating, while it is absent in the abstract parts. And their full names are also needed.

R: We have corrected “acetylation reaction.¹⁶” into “acetylation reaction ¹⁶.” We have included the following sentence in the abstract: “Combined with QM/MM computation, #A310 and #L290 were identified to control the regiospecificity of AmAT7-3 by altering the sugar positioning.” The full term of MD and QM/MM were mentioned at the beginning of the second and third paragraphs in section “Crystal structure and site-specific catalytic mechanism of AmAT7-3”, respectively.

8. Line 248: in line with our MD prediction, what is the MD prediction? Where is it?

R: We apologize for the unclear expression. Here we are referring to the MD simulation described in the second paragraph of the section “Crystal structure and site-specific catalytic mechanism of AmAT7-3”. To be clear, the sentence was revised as “The MD-predicted ratio of the two conformations was in qualitative agreement with experimental observations that the *O3'* acetylation is favored over that of the *O4'* site”.

Reviewer #2

The manuscript “Characterization, structural basis and protein engineering of a regiospecific saponin acetyltransferase from *Astragalus membranaceus*” by Wang et al. analyses the catalytic mechanism of acetylation, possible reasons for regio-selectivity in the active-site structure and substrate position and uses this information to engineer mutants with increased regio-selectivity on saponon variants. Crystal structures and mutation experiments are accompanied by modelling studies. This work contributes to our understanding of natural glycoside acylation and gives examples of engineering enzymatic regio-selectivity by single-site mutations and is therefore in principal suitable for Nature communications. There are, however, a number of points that must be addressed.

1. The authors rationalize regio-selectivity by substrate positioning. However, this is by their construction, i.e. placement of the Ac-CoA. What does the complex look like, and how do the relevant positions behave, with the substrate positioned in a way that would contradict the observed regio-selectivity? That can be answered by modelling a

negative control structure. Or there is simply no space for such a structure, which could also be mentioned/shown.

R: We thank the reviewer for reviewing our manuscript and providing insightful suggestions. The placement of Ac-CoA *via* superimposition aligns with previous reports^{1,2}. In response to the reviewer's comments, we chose a specific snapshot in which the placement of Ac-CoA appeared noticeably unfavorable for the acetylation (**Figure R1 A**). Compared with the two favorable conformations (Figure 3I and Supplementary Fig. 7), the *C-O2'/O3'/O4'* sites in this snapshot are distanced further from Cx. Consequently, in this snapshot, the reaction exhibited significantly higher barriers than that depicted in Figure 3J and K. Further, the calculated barrier for the *O3'* acetylation is much lower than that of the *O2'* acetylation, albeit that the both sites maintain a similar distance with Cx. These observations collectively emphasize that substrate positioning plays a pivotal role in determining the regio-selectivity of acetylation.

Figure R1 (A) QM(UB3LYP/B1)/MM-optimized structure of reactant complex from the selected snapshot. (B) QM(UB3LYP/B2)/MM-calculated potential energy profile (in kcal/mol) for the *O2'/O3'/O4'* acetylation in the selected snapshot. (C) The QM/MM-optimized reactive reactant complex from conformation-1 (from Figure 3I). (D) The QM/MM-optimized reactive reactant complex from conformation-2 (from Supplementary Fig. 7). All key distances in this response are given in angstrom.

2. Although packing substrates in analogy to their binding site in related complexes is often successful and the proposed structures look reasonable, the authors want to check this. A quick test is to also blindly dock Ac-CoA into the crystal structure of the protein and check if the results agree with the assumed binding site and posed. Ideally, the same is done with the known complex structure to validate the docking approach.

R: We thank the reviewer's valuable comments. Following the suggestions, we tried to dock Ac-CoA into the crystal structure. However, the binding of the docked Ac-CoA significantly deviates from the one obtained through superimposition, which seems unsuitable for the acetylation (**Figure R2 A**). Moreover, we conducted MD simulations for the docked structure, revealing that the $O2'/O3'/O4'$ sites of the substrate are distanced considerably (~ 4 - 11 Å) from the Cx site of Ac-CoA (**Figure R2 B**). These findings further indicated that the initial Ac-CoA position from docking is unreasonable for AmAT7-3. We hypothesize that due to its substantial size and flexibility, Ac-CoA can engage in intricate interactions with the protein, potentially leading to failures in docking.

The results have been integrated into the manuscript: "Since a reasonable position of Ac-CoA could not be obtained from docking (Supplementary Fig. 3), Ac-CoA was packed into the structure *via* superimposition based on the known structure of SbHCT¹⁶."

Figure R2 (A) Comparison of the docked (grey) and the superimposed (yellow) structure of Ac-CoA in AmAT7-3. (B) The fluctuation of $Cx-O2'/O3'/O4'$ distances in 100 ns MD simulation of docked structure. All dockings were performed using the AutoDock Vina tool³⁻⁴.

3. The authors report a water molecule entering between K407 and the substrate in the course of the simulation. Is this a bulk water molecule or a crystal water molecule (if there were any crystal water molecules to start with)?

R: We thank the reviewer for the comments. In the initial crystal structure, the substrate forms a direct hydrogen bond with K407 (**Figure 4A**). Nevertheless, the sugar ring is capable of rotation during MD simulations, potentially creating space for the entry of the solvent water. As a result, the introduced water is a bulk water rather than a crystal water.

This point was made clear in the manuscript: “As the sugar ring underwent rotation, it generated space for solvent water to enter”.

4. Should the two conformations, with different $Cx-O3'$ distances, not show as a bimodal distribution? Figure 3E shows a small shoulder in the $Cx-O4'$ distance distribution but nothing further that indicates two major conformations.

R: We thank the reviewer for the comments. In our study, the two conformations are discriminated based on the rotation angle of the sugar ring, rather than the bimodal distribution of distance (for details please refer to the next question). Thus, similar $Cx-O$ distances may lead to different conformations. For example, we conducted a comparison of $Cx-O3'/O4'$ distances between two snapshots extracted from conformation-1 and conformation-2, respectively (**Figure R3**). Notably, despite having similar $Cx-O4'$ distances (3.84 Å and 3.79 Å), the two snapshots belong to different conformations.

Figure R3 Comparison of $Cx-O3'/O4'$ distances (in Å) in two snapshots taken from the conformation-1 (A) and conformation-2 (B), respectively.

θ : The rotation angle of the sugar ring. The initial crystal structure is shown in grey. Multiwfn is used to fit the ring plane based on the least squares method and to compute the angle between the two planes⁵.

5. point 1: Have the two conformations been observed interchanging between each other? point 2: Or, as the text suggests, was one the starting conformation and the other one has emerged in the course of the MD simulations? point 3: In the latter case, the second conformation, the one appearing later in the simulation, is more likely to be a relaxed, equilibrated structure. point 4: Whether this is indeed the case (and with 50ns there is a chance it is not) cannot be told. The authors should therefore provide the time series of important distances in the supplementary material.

R: We thank the reviewer for the insightful comments. Following the previously reported method⁶, we distinguish two conformations using different xylosyl sugar rotation angles (**Figure R4 AB**). In conformation-1, the angle between the sugar ring simulated through MD and the sugar ring in the crystal structure was $< 75^\circ$ (as determined by Multiwfn), in which the O3' site of substrate is relatively close to the Cx of Ac-CoA. In conformation-2, the angle was $> 75^\circ$, in which the O4' site of substrate is relatively close to the Cx of Ac-CoA. The starting conformation for MD simulation was classified as conformation-1, and interchanges between conformation-1 and conformation-2 occurred frequently during the course of the 200 ns MD simulation (**Figure R4 B**). Following the reviewer's suggestion, we have provided the fluctuation of Cx-O2'/O3'/O4' distances in 200 ns MD simulation (**Figure R4 C-E**).

Description of the two conformations has been updated in the manuscript: "Due to the relatively spacious pocket of AmAT7-3, we observed significant rotational motion of the xylosyl sugar ring throughout the MD simulation. Based on the rotation angles, we segregated the substrate into two major conformations using a 75-degree threshold (Supplementary Fig. 4)". The 200-ns MD simulation result has also been included in the manuscript (Supplementary Fig. 4).

Figure R4 Fluctuation of sugar ring rotation angles and $Cx-O$ distances during MD simulations. (A) The complex structure at sugar rotation angle of 75° . The initial crystal structure is shown in grey, and the snapshot from MD simulation is presented in blue. (B) The rotation angle of the xylosyl sugar ring over a 200 ns MD simulation. A threshold angle of 75 degrees was employed to distinguish between conformation-1 and conformation-2. (C) The fluctuation of $Cx-O2'$ distances during 200 ns MD simulation. (D) The fluctuation of $Cx-O3'$ distances during 200 ns MD simulation. (E) The fluctuation of $Cx-O4'$ distances during 200 ns MD simulation.

6. H166 plays an important role in the suggested mechanism by accepting a proton from the substrate. point 1: Has the possibility of a protonated H166 been evaluated? PROPKA, which was used to find the initial protonation states or titratable residues is limited when small-molecule ligands in the vicinity have to be taken into account. point 2: Has the substrate been considered when assigning protonation states? point 3: It would also be an easy control to run a short (i.e. another 50ns) MD simulation with protonated H166 to check (and perhaps rule out) this possibility.

R: We thank the reviewer for the valuable suggestions. The protonation state of H166 is evaluated by its pK_a (~ 3.45) and its hydrogen-bonding network. The substrate was not factored into the PROPKA calculation, but it was considered in the QM/MM simulations. Following the comment of referee, MD simulation with the doubly protonated H166 was also performed. In this scenario, the doubly protonated H166 tends to form a H -bond with either $O3'$ or $O4'$ of substrate (**Figure R5**), with no available base to mediate the reaction. Clearly, the notion of H166 being doubly protonated is implausible.

Figure R5 (A) The distance fluctuation between the doubly protonated H166 and O4'. (B) The distance fluctuation between the doubly protonated H166 and O3'. (C) An illustrative snapshot in which H166 forms an H-bond with O4'. (D) An illustrative snapshot in which H166 forms an H-bond with O3'.

7. Among the many mutants there is no H166X mutation. Would that not be a good control of the importance of H166 (if not of its exact role)?

R: We thank the reviewer for the suggestions. The mutants of H166 were generated by substituting His to Gly (uncharged), Asp (acidic), Val (non-polar, aliphatic), Arg (basic). Functions of the mutants were characterized using purified proteins following the same procedure as the wild-type. Astragaloside IV (**1**) and Ac-CoA served as the substrate and acyl donor, respectively. However, none of the above-mentioned mutants exhibited the ability to acetylate the substrate (**Figure R6**). This result further validates the critical role of H166 in the function of AmAT7-3.

The results have been added into the manuscript: In addition to the established critical role of H166 (Supplementary Fig.8), ten amino acids within 5 Å from the substrate were found to participate in building the internal structure of the active cavity. **Figure R6** has been added into the manuscript as Supplementary Fig. 8.

Figure R6 The catalytic activity of H166 mutants.

The reactions were carried out using Astragaloside IV (**1**) as the substrate and Ac-CoA as the acyl donor. The reaction condition was at pH 6.0 (50 mM NaH₂PO₄-Na₂HPO₄), incubated at 30°C for 30 min. "Reaction" refers to the acetylation reaction catalyzed by H166 mutants, while the "Control" involved the use of boiled enzyme within the reaction mixture. LC/MS extracted ion chromatograms were presented by extracting the ions for the substrates and the mono-acetylated products: [M-H+HCOOH]⁻ (*m/z* 829.45), [M+Ac-H+HCOOH]⁻ (*m/z* 871.46).

8. A310F shows a preference for C4' acetylation, similar to the A310W mutant. This is quite likely for sterical reasons as discussed for A310W. Similar effects are also

observed when mutating A312 which has a different location and might “push” in another direction. It would be worth making a model of those mutants that show the sterical demands (no reaction calculation required).

R: We thank the reviewer for the insightful suggestions. In order to assess whether these mutants alter regioselectivity through steric effects, we conducted MD simulations for A312F/W and A310F. The results revealed that they all exhibit steric effects (distance $<3 \text{ \AA}$) similar to A310W. For instance, the minimum atomic distance between the sugar ring and W312 remains at approximately 3 \AA , indicating the presence of steric influence (**Figure R7**). Similar to A310W, A312W induces a clockwise rotation of the sugar ring due to steric hindrance, resulting in the proximity of the *O4'* site to *Cx* for acetylation. Consequently, these mutants exhibit comparable steric effects that impact their regioselectivity.

These findings have been incorporated into the manuscript: “Similar to A310W, steric effects were observed for A310F, A312W and A312F variants, as demonstrated by MD simulations and mutagenesis results (Supplementary Fig. 11)”. **Figure R7** has been included in the manuscript as Supplementary Fig. 11.

Figure R7 (A) The fluctuation of minimal distance between the sugar ring and key residues in the mutants. (B) Representative snapshots of the reactant complexes for the mutants.

9. All the other mutations that have an effect on activity, rather than on regio-selectivity are very interesting. For example, mutating A310 to R or D does not make much of a difference. D311 can be mutated to R without apparent consequences, although one would assume the hydrogen-bonding pattern, and thus the binding mode of the substrate, to change significantly. Without further (modelled) structures, this will be a bit speculative but perhaps the authors can discuss more of the mutation effects on activity based on their reaction calculations.

R: We thank the reviewer for the suggestions. Different mutants employ distinct mechanisms to influence protein activity due to the intricate nature of the enzymatic

system. Here we focus on A310R and D311R mutants to discuss the mutation effects on activity. When compared to the wild type, A310R and D311R selectively catalyze the *O3'* acetylation of the substrate. For both mutants, we observed that the introduced residue R can form a persistent hydrogen bond with *O2'* of the substrate (distance within 3 Å). This interaction impedes the rotation of the sugar ring and stabilizes the substrate within conformation-1 (**Figure R8**). Consequently, the reaction is confined to *O3'* acetylation, aligning with our experimental observations. In terms of activity, the conversion rates of AmAT7-3, AmAT7-3_{A310R} and AmAT7-3_{D311R} were 85.3% (57.7% *O3'*), 48.2% (all *O3'*), and 49.6% (all *O3'*). Slight changes in the *O3'* acetylation rate should be attributed to the altered position of the sugar ring.

These findings have been added into the manuscript: “Meanwhile, A310R and D311R could form hydrogen bonds with *O2'*, effectively stabilizing the substrate and restricting the reaction to *O3'* acetylation (Supplementary Fig. 11)”. **Figure R8** has been included in the manuscript as Supplementary Fig. 11.

Figure R8 (A) Fluctuation in the rotation angle of the sugar ring in A310R. (B) Fluctuation in distance between the *N* atom of R310 and *O2'* of the sugar ring of the substrate in A310R mutant. (C) A representative snapshot showing the binding conformation of the sugar ring in the A310R mutant. (D) Fluctuation in the rotation angle of the sugar ring in D311R. (E) Fluctuation in distance between the *N* atom of R311 and *O2'* of the sugar of the substrate in D311R mutant. (F) A representative snapshot showing the binding conformation of the sugar ring in the D311R mutant.

10. The classical MD simulations, performed prior to the QM/MM calculations already suggest a conformational flexibility of the substrate. The authors tried to take this into account by modelling the reaction starting from two different conformations. Their optimization-only approach, however, does not allow conformational changes in the course of the reaction or does it account for substrate or protein flexibility and possible entropic effects associated therewith. This should at least be discussed (since QM/MM free energy calculations are far more demanding than the optimizations used in the present work and more modelling/simulations are needed for testing binding poses, protonation states and consequences for the mechanism already).

R: We thank the reviewer for the instructive suggestions. Unlike bimolecular process (substrate entry or product dissociation), the monomolecular catalytic reaction usually involves minor entropy contribution. For example, our previous study⁷ shows that H-abstraction by Fe(IV)=O species in non-heme FtO has an electronic barrier of 17.7 kcal/mol, while the frequency analysis shows that the zero-point energy (ZPE) correction will reduce the barrier by 2.6 kcal/mol, whereas the molecular entropy correction ($-T\Delta S$) will increase the barrier by 1.6 kcal/mol. The net effect is a close match between the computed electronic energy barrier of 17.7 kcal/mol and the computed free energy barrier of 16.1 kcal/mol. Such a close match was proven by Thiel and co-workers, who carried out careful sampling on a variety of H-abstraction processes in P450⁸⁻⁹. Indeed, the potential energy approximation has been so extensively used for the study of enzymatic catalysis¹⁰⁻¹⁷.

Indeed, even for QM/MM-MD based free energy calculations, it is still not able to sample the conformational change of protein environment, this is mainly because that the time scale of QM/MM MD simulations (~ 10 ps) is much smaller than those of the conformational switch of substrate (>1 ns) or the conformational change of protein environment ($>>1$ ns).

Following the comment of the reviewer, we have added explanations in the manuscript: “According to our previous and other groups’ work, entropy correction usually plays minor effects in the catalytic reactions of enzymes⁶³⁻⁶⁵. Given the high computational cost for the QM/MM free energy calculations, we use the electronic energy barriers as estimates of the free energy barriers in the enzyme, which was demonstrated to be practical in previous work⁶⁶⁻⁶⁸.”

11. There are also some details missing in the methods section of the computational studies: How many steps, using which optimisation algorithm were used for minimisation, equilibration and annealing? Which thermostat was employed and which barostat (if applicable) to control which temperature (and pressure)?

R: Following the suggestions of the reviewer, we have added these technical details in method part: The complex system first was subjected to 5000 steps of steepest descent and 5000 steps of conjugate gradient minimization, with the protein held fixed by using position restraints with a force constant of $500 \text{ kcal}\cdot\text{mol}^{-1}\text{\AA}^{-2}$. And an additional 5000 steps of steepest descent and 25000 steps of conjugate gradient minimization was performed to fully optimize the system without restraints. Then, the systems were annealed from 0 to 300 K for 50 ps with the NVT ensemble, during which the constraint of 15 kcal/mol/\AA was applied. Then, the density equilibration for 1.0 ns was conducted under the NPT ensemble to obtain a uniform density, where the target temperature of 300 K was kept with the Langevin thermostat¹⁸ and a 2 ps collision frequency, and the 1.0 atm target pressure was maintained with the Berendsen barostat¹⁹ and a pressure relaxation time of 1 ps. Subsequently, all the restraints on the complex systems were removed, and the enzyme complexes were equilibrated for 4 ns under the NPT ensemble. Lastly, a productive MD simulation of 200 ns was carried out under the NPT ensemble. During the simulations, the covalent bonds involving hydrogen atoms were constrained with the SHAKE method²⁰, and the integration step was set to 2 fs. A cutoff radius of 8 Å was set for nonbonded interactions, while the long-range electrostatic interactions were treated using the Particle Mesh Ewald (PME) method²¹. All MD simulations were performed using Amber 18 software package²². The above methods have been added into the “Classical MD simulations” section of the manuscript.

12. What are “major conformations observed in the trajectory”?, How were “representative snapshots” chosen, i.e. in which respect are they representative?

R: We thank the reviewer for the insightful comments, and apologize for the unclear expression. “The major conformations” represents conformation-1 and conformation-2, which constitute two sets of conformations differentiated by distinct sugar rotation angles, partitioned at a 75° threshold (for details please refer to question 5). Numerous snapshots are associated with each conformation. The representative snapshot was a most reactive instance featuring the closest Cx-O distances from the near-attack conformation in either conformation-1 or conformation-2 trajectory, as shown in **Figure R9**. Snapshots with the close Cx-O distances were chosen for QM/MM calculation, as our calculations have demonstrated that snapshots with extended Cx-O2'/O3'/O4' distances require significantly higher barriers for acetylation (data not shown). In the revised manuscript, we used “most reactive snapshots” instead of “representative snapshots”.

The division of the two conformations in manuscript have been revised into: “Due to the relatively spacious pocket of AmAT7-3, we observed significant rotational motion of the xylosyl sugar ring throughout the MD simulation. Based on the rotation angles, we segregated the substrate into two major conformations using a 75-degree threshold (Supplementary Fig. 4)”. Please also refer to the answer to question 5 of Reviewer #2.

The “representative snapshots” has been replaced by “most reactive snapshots”, and was explained in the manuscript as: ...“QM/MM computation was conducted on two most reactive snapshots featuring close Cx-O3' or Cx-O4' distance from the near-attack conformation in either conformation-1 or conformation-2 (Supplementary Fig. 5-6).” Supplementary Fig. 5 is identical to **Figure R9**.

Figure R9 (A) A most reactive snapshot in conformation-1 used in QM/MM calculations (the same snapshot as the one in Figure 3G). (B) A most reactive snapshot

in conformation-2 used in QM/MM calculations (the same snapshot as the one in Figure 3H).

13. Please provide the exact composition of the QM region, i.e. list all residues and positions of link atom placement.

R: Following the suggestion of the referee, we have plotted the QM region (in blue) for the complexes of AmAT7-3, AmAT7-3_{A310G} and AmAT7-3_{A310W}, respectively. The red line delineates the boundary between the QM region and the MM region (**Figure R10**). This Figure has been added to the manuscript (Supplementary Fig. 6).

Figure R10 (A) The QM region (in blue) used for AmAT7-3. (B) The QM region (in blue) used for AmAT7-3_{A310W}. (C) The QM region (in blue) used for AmAT7-3_{A310G}.

The red line delineates the boundary between the QM region and the MM region.

14. Which optimisation algorithm was use and what were the convergence criteria? How were transition states searched (i.e. guessing transition structures, potential energy scan, chain-of-state optimisation) and which algorithm was used for that?

R: The limited memory quasi-Newton (L-BFGS) algorithm in the DL-FIND optimizer was used in the QM/MM geometry optimization in Chemshell, in which the default convergence criteria has been used. The transition states were determined as the highest point on the potential energy surface, which is further optimized using the P-RFO

optimizer. All these technical details have been added into the method part “QM/MM calculation” in the revised manuscript. Again, we thank Reviewer #2 for carefully reviewing our manuscript and providing insightful comments.

Reviewer #3

In this manuscript, the authors report the structure and function analysis of saponin acetyltransferase from *Astragalus membranaceus*. The acetyltransferase AmAT7-3 which catalyzes acetylation of C3'/C4' position of astragaloside IV was identified by genome mining method. The crystal structure, MD simulation, and QM/MM analysis of AmAT7-3 revealed the key active site residues for regioselectivity and the two major conformations of substrate in the active site. Furthermore, the authors successfully altered the regioselectivity by mutagenesis experiments. The authors also demonstrated the generation of acetylated compounds by using various substrate analogs. This manuscript is well-written and easy to follow. The detailed structural and calculation analysis of the enzyme provide detailed information on regioselectivity of acetylation reaction. The results shown in this manuscript would attract the interests of the researchers in the field of natural product chemistry, biosynthesis, and enzymology. Therefore, I would recommend it for acceptance with some revisions.

1. To further analysis of the acetyl migration, time course reaction should be performed.

R: We thank the reviewer for the positive comments and constructive suggestions. We have added new data to show the time course for the acetyl migration as in **Figure R11**. The following discussion has been added:

“To better understand the spontaneous acetyl migration reaction, the time course was obtained under 50 mM Na₂HPO₄-NaH₂PO₄ at pH 7.6, using **1a/1b/1d** (C3'/C4'/C2'-OAc) as the substrate. Interestingly, both **1a** and **1b** were converted to **1d** as a main acetylated product after 8 h (58.7-70.2% within the acetylated products), and only a small portion of **1d** was converted back to **1a/1b** (<25%) (Figure 2 D-F). These results indicated that **1a/1b**, the major products for AmAT7-3, could be converted into the dominant saponin **1d** in *A. membranaceus* through acetyl migration.” Figure 2 D-F are identical to **Figure R11**.

Figure R11 Time-course of the acetyl migration for **1a**, **1b**, and **1d**, respectively.

2. The authors should comment on the relationship between the ratio of conformation-1/conformation-2 and the ratio of *O3'*-/*O4'*-products.

R: We thank the reviewer for the instructive comments. To evaluate the ratio of conformation-1/conformation-2 and *O3'*-/*O4'*-products, we have further categorized the reactive snapshots based on our QM/MM optimization outcomes and other reports²³. Concerning *O3'* acetylation, the *Cx-O3'* distance of less than 3.4 Å is required, alongside a nucleophilic attack angle (formed by the connection between the oxygen on the hydroxyl group of the sugar and the carbonyl group of Ac-CoA) ranging from 90° to 120°. For *O4'* acetylation, the *Cx-O4'* distance of less than 3.4 Å is necessary, accompanied by a nucleophilic attack angle spanning from 80° to 110°. According to these criteria, we observed a reactive snapshot distribution of 43:4 between conformation-1 and conformation-2 within a 50-ns MD simulation. The ratio of *O3'*-acetylated /*O4'*-acetylated products is approximately 3.8:1 at 15 min, according to Figure 1F. The computational results are in qualitative agreement with experimental observations, underscoring the prevalence of *O3'* acetylation as the major product. This comment has been added into the manuscript as: “The MD-predicted ratio of the two conformations was in qualitative agreement with experimental observations that the *O3'* acetylation is favored over that of the *O4'* site”. Detailed discussions were included in the legend of Supplementary Fig. 4.

3. The sequence comparison among AmAT7-3 and homologues around L290 region should also be shown.

R: We thank the reviewer for the valuable comments. We have aligned all residues forming the active pocket (#29-#34, #372-#377, #284-#291, #309-#318, #359-#365) with reported glycoside acetyltransferases. However, except for #310-314, no other conserved or distinct motifs were observed at these sites, despite most of the residues being aliphatic (L/A/P/I/V) at #289-290 (**Figure R12**).

The results have been added to the manuscript: “To demonstrate the sequence and structure specificity for AmAT7-3, we aligned its protein sequence with six reported glycoside acetyltransferases (Fig. 4H, Supplementary Fig. 15). ... We also aligned the other residues forming the active pocket (#29-#34, #372-#377, #284-#291, #309-#318, #359-#365) with reported glycoside ATs (Supplementary Fig. 15). However, conserved or distinct motifs were not observed at these sites, despite most of the residues being aliphatic (L/A/P/I/V) at #289-290.” Supplementary Fig. 15 is identical to **Figure R12**.

Figure R12 Sequence analysis for AmAT7-3 and other flavonoid and saponin acetyltransferases.

4. The key HMBC correlations between acetyl group and sugar are not clear in some NMR figures. The closed-up views should be added.

R: We thank the reviewer’s suggestion. We have carefully checked the readability of all NMR figures, and did our best to improve the resolution. The closed-up views of HMBC and HSQC correlations were provided for Supplementary Fig. 36 (**3b**),

Supplementary Fig. 48 and 49 (7a). Supplementary Fig. 24/28/37 have been adjusted to be clear. Updated figures were included in the revised Supporting information.

Supplementary Fig. 36 HMBC spectrum of **3b** in pyridine- d_5 (400 MHz).

Supplementary Fig. 48 HSQC spectrum of **7a** in $\text{pyridine-}d_5$ (400 MHz).

Supplementary Fig. 49 HMBC spectrum of **7a** in pyridine-*d*₅ (400 MHz).

5. Interactions between aglycon and active site residues should be more discussed. In addition, the authors should explain the structural basis for the broad substrate specificity of AmAT7-3 toward various aglycons.

R: We thank the reviewer for the insightful suggestion. To elucidate the interactions between aglycon and the active site, the ligand interaction pattern of substrate **1** was explored using Schrödinger using default parameters in ligand interaction (**Figure R13 A**). The prevailing interactions between the substrate and the protein involve hydrophobic contacts and electrostatic effects. Notably, numerous residues within the active pocket engage in hydrophobic interactions with the substrate. Among these residues, M364, P360, L372, I288, Y286, and I316 were around the aglycon. Additionally, I374 and A376 are oriented toward the xylosyl moiety, while V314, A312,

and W353 are in proximity to the glucosyl group. Furthermore, electrostatic interactions were observed on R349/D311 with the glucosyl group, as well as K407/R38 with the xylosyl group. Interactions between aglycon and the active site residues were partly supported by our mutagenesis study involving I288 (Figure 4B).

To explain the structural basis for the broad substrate specificity of AmAT7-3, we docked representative substrates ziyuglycoside II (**3**), bufalin 3-*O*- β -D-glucoside (**4**), 20(*R*)-ginsenoside Rh1 (**6**) and glycyrrhetic acid 3-*O*- β -D-glucuronide (**7**) into the crystal structure of AmAT7-3 to explore their interaction patterns (Figure R13 B-E). Interestingly, while the amino acid residues that interact with different substrates may overlap, they might serve distinct roles with different positions of each substrate. For example, A310 was close to the sugar ring of substrates **1/4**, yet adjacent to the aglycone of substrates **3/6/7**. Similar observations hold for I288 and L290, which are situated close to the aglycone of substrates **1/6/7**, while close to the sugar ring of substrates **3/4**. Residues R38 and E39 appear conserved in interacting with the hydroxyl groups on the sugar moiety. However, the specific interaction sites vary across different substrates. The analysis of the interaction patterns above indicated that structural basis for the substrate promiscuity arises from the large volume of the active pocket, which can readily accommodate sizable saponin molecules. The orientation of the saponin molecules may be determined by conserved amino acids such as R38 and E39, which establish hydrogen bonds with the sugar moiety. To support this hypothesis, we introduced mutations at three positions, R38, E39, and A310. The results indicated that all variants exhibited an impact on reaction activity or selectivity (Figure R13 F). In comparison to the wild-type, the overall conversion rates of all mutants decreased, with the exception of E39G with substrate **7** (where E39G potentially enhanced activity through precise modulation of the sugar ring's position).

These findings have been briefly integrated into the manuscript: “Based on the examination of ligand interaction patterns, the structural basis behind substrate promiscuity likely stems from the large volume of the active pocket, facilitating the accommodation of sizable saponin molecules. The positioning of the saponins might be determined by conserved amino acids such as R38 and E39, which establish hydrogen bonds with the sugar moiety (Supplementary Fig. 54)”. Detailed discussions were included in the legend of Supplementary Fig. 54, which was identical to Figure R13.

Meanwhile, we believe that the mechanism behind substrate promiscuity extends beyond the scope of the story presented in our paper. It is a topic that merits future

exploration, particularly in the context of engineering of AmAT7-3 to achieve highly specific acetylation of natural saponins.

6. The *Fo-Fc* omit map for ligand in the crystal structure need to be added in SI.

R: We thank the reviewer for the suggestion. The *Fo-Fc* omit map for ligand in the crystal structure were added in Supplementary Fig. 55. The figure was generated in Pymol.

Supplementary Fig. 55 The *Fo-Fc* omit map for ligand in the crystal structures.

References

1. Tezé, D., Shuoker, B., Chaberski, E., Kunstmann, S., Fredslund, F., Nielsen, T., Stender, E., Peters, G., Nordberg Karlsson, E., Welner, D., & Abou Hachem, M. The catalytic acid-base in gh109 resides in a conserved gghgg loop and allows for comparable alpha-retaining and beta-inverting activity in an n-acetylgalactosaminidase from akkermansia muciniphila. *ACS Catal* **10**, 3809–3819 (2020).
2. Wang, X., Li, Y., Chen, X., Zhou, Z., & Yao, J. Human acetyl-coA carboxylase 1 is an isomerase: carboxyl transfer is activated by catalytic effect of isomerization. *J Phys Chem B* **123**, 6757–6764 (2019).
3. Trott, O., & Olson, A. AutoDock Vina: improving the speed and accuracy of docking with a new scoring function, efficient optimization, and multithreading. *J Comput Chem* **31**, 455–461 (2010).
4. Pettersen, E., Goddard, T., Huang, C., Couch, G., Greenblatt, D., Meng, E., & Ferrin, T. UCSF Chimera--a visualization system for exploratory research and analysis. *J Comput Chem* **25**, 1605–1612 (2004).
5. Tian L., Feiwu C., Multiwfn: a multifunctional wavefunction analyzer. *J Comput Chem* **33**, 580-592 (2012).
6. Kato, A., Hayashi, E., Miyauchi, S., Adachi, I., Imahori, T., Natori, Y., Yoshimura, Y., Nash, R., Shimaoka, H., Nakagome, I., Koseki, J., Hirono, S., & Takahata, H. α -1-C-butyl-1,4-dideoxy-1,4-imino-l-arabinitol as a second-generation iminosugar-based oral α -glucosidase inhibitor for improving postprandial hyperglycemia. *J Med Chem* **55**, 10347–10362 (2012).
7. Wang, B., Cao, Z., Sharon, D., & Shaik, S. Computations reveal a rich mechanistic variation of demethylation of *N*-methylated DNA/RNA nucleotides by FTO. *ACS Catal* **5**, 7077-7090 (2015).
8. Senn, H., Thiel, S., & Thiel, W. Enzymatic hydroxylation in *p*-hydroxybenzoate hydroxylase: a case study for QM/MM molecular dynamics. *J Chem Theory Comput* **1**, 494–505 (2005).
9. Hans, M., Johannes, K., Jurgen, B., Walter, T., Finite-temperature effects in enzymatic reactions-insights from QM/MM free-energy simulations. *Can J Chem* **87**, 1322–1337 (2009).
10. Lee, Y., Hou, X., Chen, R., Feng, J., Liu, X., Ruszczycky, M., Gao, J., Wang, B., Zhou, J., & Liu, H. Radical *S*-adenosyl methionine enzyme BlsE catalyzes a radical-mediated 1,2-Diol dehydration during the biosynthesis of blasticidin S. *J Am Chem Soc* **144**, 4478–4486 (2022).
11. Deng, W., Lu, Y., & Liao, R. Revealing the mechanism of isethionate sulfite-lyase by QM/MM Calculations. *J Chem Inf Model* **61**, 5871–5882 (2021).
12. Chaturvedi, S., Ramanan, R., Hu, J., Hausinger, R., & Christov, C. Atomic and electronic structure determinants distinguish between ethylene formation and l-arginine hydroxylation reaction mechanisms in the ethylene-forming enzyme. *ACS Catal* **11**, 1578-1592 (2021).

13. Peng W., Yan S., Zhang X., Liao L., Zhang J., Shaik S., Wang B. How do preorganized electric fields function in catalytic cycles? The case of the enzyme tyrosine hydroxylase. *J Am Chem Soc* **44**, 20484–20494 (2022).
14. Zhang X., Wang Z., Li Z., Shaik S., Wang B. [4Fe–4S]-Mediated proton-coupled electron transfer enables the efficient degradation of chloroalkenes by reductive dehalogenases. *ACS Catal* **13**, 1173–1185 (2023).
15. Ryde U. QM/MM Calculations on proteins. *Methods Enzymol* **577**, 119–158 (2016).
16. Shaik, S., Cohen, S., Wang, Y., Chen, H., Kumar, D., & Thiel, W. P450 enzymes: their structure, reactivity, and selectivity-modeled by QM/MM calculations. *Chem Rev* **110**, 949–1017 (2010).
17. Senn, H., & Thiel, W. QM/MM methods for biomolecular systems. *Angew Chem Int Ed* **48**, 1198–1229 (2009).
18. Izaguirre, J., Catarello, D., Wozniak, J., & Skeel, R. Langevin stabilization of molecular dynamics. *J Chem Phys* **114**, 2090-2098 (2001).
19. Berendsen, H., Postma, J., Gunsteren, W., Dinola, A., & Haak, J. Molecular-dynamics with coupling to an external bath. *J Chem Phys* **81**, 3684 (1984).
20. KrÄUutler, V., Gunsteren, W., & Philippe H. A fast shake algorithm to solve distance constraint equations for small molecules in molecular dynamics simulations. *J Comput Chem* **22**, 501-508 (2001).
21. Darden, T., York, D., & Pedersen, L. Particle mesh Ewald: An $N \cdot \log(N)$ method for Ewald sums in large systems. *J Chem Phys* **98**, 10089-10092 (1993).
22. Case, D., Ben-Shalom, I., Brozell, S., Cerutti, D., Cheatham, T., III, Cruzeiro, V., Darden, T., Duke, R., Ghoreishi, D., Gilson, M, Gohlke, H., Goetz, A., Greene, D., Harris, R., Homeyer, N., Huang, Y., Izadi, S., Kovalenko, A., Kurtzman, T., Lee, T. S., LeGrand, S., Li, P., Lin, C., Liu, J., Luchko, T., Luo, R., Mermelstein, D., Merz, K., Miao, Y., Monard, G., Nguyen, C., Nguyen, H., Omelyan, I., Onufriev, A., Pan, F., Qi, R., Roe, D., Roitberg, A., Sagui, C., Schott-Verdugo, S., Shen, J., Simmerling, C., Smith, J., Salomon-Ferrer, R., Swails, J., Walker, R., Wang, J., Wei, H., Wolf, R., Wu, X., Xiao, L., York, D., Kollman, P. AMBER, 2018, University of California: San Francisco, 2018.
23. Sadiq, S., & Coveney, P. Computing the role of near attack conformations in an enzyme-catalyzed nucleophilic bimolecular reaction. *J Chem Theory Comput* **11**, 316–324 (2015).

Reviewers' Comments:

Reviewer #2:

Remarks to the Author:

The authors have substantially revised and their manuscript and addressed all my concerns. there are only a few minor points (see below) that need attention before the manuscript can be published.

I am not sure "a most reactive snapshot" is appropriate if there is more than one of those. Perhaps "a reactive snapshot" is enough.

The authors added to the caption of Figure 1E: "Data are presented in mean \pm SD ($n \geq 2$ independent experiments)." However, there are no error bars visible in this figure. If these are too small to be seen over the markers, the authors want to decrease the marker size (or omit them completely) in favour of the error bars.

The same holds for figure 2F.

The description "The transition states were determined as the highest point on the potential energy surface" (line 601) is likely not correct, since a transition state is a first-order saddle point on the potential energy surface, not a maximum.

Reviewer #3:

Remarks to the Author:

The authors have addressed this reviewers comments satisfactorily and the manuscript should be accepted for publication.

Response to reviewers' comments

Reviewer #2

The authors have substantially revised their manuscript and addressed all my concerns. There are only a few minor points (see below) that need attention before the manuscript can be published.

1. I am not sure "a most reactive snapshot" is appropriate if there is more than one of those. Perhaps "a reactive snapshot" is enough.

R: We thank the reviewer for reviewing our manuscript again carefully. As suggested by the reviewer, we have corrected the "a most reactive snapshot" into "a reactive snapshot".

2. The authors added to the caption of Figure 1E: "Data are presented in mean \pm SD ($n \geq 2$ independent experiments)." However, there are no error bars visible in this figure. If these are too small to be seen over the markers, the authors want to decrease the marker size (or omit them completely) in favor of the error bars. The same holds for figure 2F.

R: We thank the reviewer for the valuable comments. These error bars are too small to see so we decreased the marker size of Figure 2F to make them clear.

Figure R1 Time-course of the acetyl migration for **1d**.

3. The description "The transition states were determined as the highest point on the potential energy surface" (line 601) is likely not correct, since a transition state is a first-order saddle point on the potential energy surface, not a maximum.

R: We thank the reviewer for the comments and apologize for the incorrect description. We have revised this sentence as "The transition states were determined as the highest energy structure from the adiabatic scan".

Reviewer #3

The authors have addressed this reviewers comments satisfactorily and the manuscript should be accepted for publication.

R: We thank Reviewer #3 for reviewing our manuscript again and giving positive comments.